# Deep Unsupervised Domain Adaptation for Time Series Classification: a Benchmark

## Abstract

Unsupervised Domain Adaptation (UDA) aims to harness labeled source data to train models for unlabeled target data. Despite extensive research in domains like computer vision and natural language processing, UDA remains underexplored for time series data, which has widespread real-world applications ranging from medicine and manufacturing to earth observation and human activity recognition. Our paper addresses this gap by introducing a comprehensive benchmark for evaluating UDA techniques for time series classification, with a focus on deep learning methods. We provide seven new benchmark datasets covering various domain shifts and temporal dynamics, facilitating fair and standardized UDA method assessments with state of the art neural network backbones (e.g. Inception) for time series data. This benchmark offers insights into the strengths and limitations of the evaluated approaches while preserving the unsupervised nature of domain adaptation, making it directly applicable to practical problems. Our paper serves as a vital resource for researchers and practitioners, advancing domain adaptation solutions for time series data and fostering innovation in this critical field.

## 1 Introduction

The realm of time series classification has witnessed a remarkable surge in recent years, driven by the omnipresence of time series data across multitude of domains. These temporal data sequences, characterized by their inherent ordering, have become central in diverse cognitive tasks. Time Series Classification (TSC) stands as a pivotal task within this domain, entailing the precise labeling of time series data given a set of labeled training examples (Ismail Fawaz et al., 2019). With the advent of the Internet of Things (IoT) and the proliferation of big data, the manipulation and analysis of time series data have become integral components of data science practices spanning fields such as medicine, telecommunications, remote sensing, and human activity recognition (Sanz et al., 2022; Kamalov et al., 2021). This paper embarks on a comprehensive exploration of the intricate landscape of TSC, offering novel insights, methodologies, and applications within this evolving discipline.

In numerous scenarios, including those devoid of temporal dynamics, a notable discrepancy arises between the data observed in the deployment environment of a machine learning model and the data employed during its training phase. This disparity emerges due to the distinct characteristics inherent to the new operational context. For instance, variations in genomic profiles or the medical condition of a patient (Wagner et al., 2020), fluctuations in the density of devices or shifts in patterns of communication data utilization (Park & Simeone, 2022), or changes in weather conditions and soil properties within remote sensing applications (Nyborg et al., 2022), all contribute to this phenomenon. An analogous situation occurs when models are trained using synthetic data then later used for real-world data analysis. Consequently, it becomes imperative to adapt the model to accommodate this data distribution shift, which constitutes the fundamental objective of domain adaptation. This challenge assumes an even greater degree of complexity when labels are entirely absent in the new domain, thereby giving rise to the subfield of unsupervised domain adaptation.

Domain Adaptation (DA) and Unsupervised Domain Adaptation (UDA) have garnered substantial attention in the fields of Natural Language Processing (NLP) and Computer Vision (CV) (Ruder et al., 2019; Li et al., 2020; Xu et al., 2022). These domains have witnessed the emergence of numerous methodological approaches to address data distribution disparities. One commonly employed strategy involves the transfer of specific layers from Neural Networks (NN), followed by the

freezing of these layers during the forward pass (Ismail Fawaz et al., 2018). In the context of data shift, particularly covariate shift, more intricate algorithms have been developed. These include: (1) Domain-Adversarial Neural Network (DANN) (Ganin et al., 2016), which advocates adversarial training to minimize the divergence between source and target domain distributions; (2) Conditional Adversarial Domain Adaptation (CDAN) (Long et al., 2018), which aligns conditional distributions; and (3) algorithms grounded in optimal transport theory (Courty et al., 2015; Damodaran et al., 2018). To assess the efficacy of these methodologies, benchmarking initiatives have been established (Zhao et al., 2020). Ringwald & Stiefelhagen (2021) conducted a comparative evaluation of these approaches using image datasets, revealing notable challenges related to image noise and annotation errors while demonstrating significant improvements upon addressing them. Zhao et al. (2020) concentrated predominantly on evaluating deep unsupervised methods within their benchmarking framework. While this benchmark tackles classification tasks, other works focus on DA regression tasks, such as Wei et al. (2022) who proposed to take advantage of a Kernel that takes into consideration the domain of the compared samples.

In contrast to the well-established benchmarking frameworks in computer vision and NLP, the time series data community currently lacks a comprehensive overview of the existing techniques and datasets available for benchmarking UDA in the context of TSC, which is highlighted in a recent UDA method for temporal data called SASA (Cai et al., 2021). While an initial work in this direction has been made by Ragab et al. (2023), it remains somewhat constrained in terms of data diversity, dataset quantity, and statistical analysis for assessing the performance of multiple time series classifiers across various datasets. Unlike the review in Ragab et al. (2023), the benchmark we introduce is specifically tailored to deep learning approaches, for unsupervised domain adaptation in the realm of time series data. We evaluate 9 algorithms integrated with cutting-edge backbone architectures (e.g. InceptionTime's backbone) and scrutinize their performance across a set of 12 datasets, including 7 novel ones we introduce to diversify domain contexts. Additionally, we delve into the crucial aspect of hyperparameter tuning criteria, an important point in UDA, given the absence of labeled data in the target domain.

## 2 Unsupervised Domain Adaptation Background

### 2.1 Notations and Problem Statement

Let $D_S = \{(\mathbf{X}_i^s, \mathbf{y}_i^s)\}_{i=1}^{n_S}$ be a source domain consisting of $n_S$ labeled time series, where $\mathbf{X}_i^s$ represents the time series data and $\mathbf{y}_i^s$ represents the corresponding labels. Additionally, let $D_T = \{\mathbf{X}_j^T\}_{j=1}^{n_T}$ represent a target domain solely consisting of $n_T$ unlabeled time series. The objective of UDA is to learn a classifier capable of accurately estimating the labels of the time series in the target domain by making use of the labeled time series from source domain. As some datasets contain multiple domains, each adaptation from one domain to another is referred to as a *scenario*.

In the majority of the DA literature, the theoretical findings (Ben-David et al., 2006) as well as the underlying motivation of the algorithms (Ganin et al., 2016) presume that the shift between the two domains adheres to the covariate shift assumption (Farahani et al., 2021). The latter posits that the distribution of input data (i.e., the time series data) differs between the source and target domains, while the conditional distribution of labels, given the input data, remains unchanged, i.e. $p_S(\mathbf{X}) \neq p_T(\mathbf{X})$ but $p_S(\mathbf{y}|\mathbf{X}) = p_T(\mathbf{y}|\mathbf{X})$ for any $\mathbf{X} \sim \mathcal{X}$ and $\mathbf{y} \sim \mathcal{Y}$, with $\mathcal{X}$ and $\mathcal{Y}$ the input and label spaces, and $p_S$ and $p_T$ the source and target distributions.

### 2.2 UDA Algorithms for time series

The focus of this work is on deep UDA algorithms for TSC, where the goal is to classify unlabeled data in the target domain. These algorithms rely on two main components: a backbone, which encodes the input into a domain-invariant latent space, and a classifier. During training, labeled data from the source domain is used to train the classifier for the main classification task, while unlabeled data from the target domain is used to adapt the model to the target domain. Table 1 summarizes the algorithms considered in this work. In the following, we describe these algorithms in more detail.

**Baseline** InceptionTime (Ismail Fawaz et al., 2020) is used as a baseline with no adaptation to compare with UDA approaches and highlight the benefits of applying domain adaptation. This

Table 1: Deep UDA algorithms for TSC (top) and baselines (bottom). All algorithms consist of a backbone and a classifier, other modules are listed in the table. $\mathcal{L}_C$, $\mathcal{L}_A$, $\mathcal{L}_{\text{VRNN}}$, $\mathcal{L}_R$, $\mathcal{L}_{\text{Contrastive}}$, $H$, and $\mathcal{L}_{\text{Sinkhorn}}$ denote the classification, adversarial, VRRN, reconstruction, and contrastive loss functions, the entropy and the Sinkhorn divergence respectively. See A.3 for more details.

| Algorithms | Backbone | Other Modules | Loss function |
|---|---|---|---|
| VRADA | VRNN | Discriminator | $\mathcal{L}_C + \mathcal{L}_A + \mathcal{L}_{\text{VRNN}}$ |
| CoDATS | 1D CNN | Discriminator | $\mathcal{L}_C + \mathcal{L}_A$ |
| InceptionDANN | Inception | Discriminator | $\mathcal{L}_C + \mathcal{L}_A$ |
| InceptionCDAN | Inception | Discriminator, Multilinear Map | $\mathcal{L}_C + \mathcal{L}_A$ |
| CoTMix | 1D CNN | Temporal Mixup | $\mathcal{L}_C + \mathcal{L}_{\text{Contrastive}} + H$ |
| InceptionMix | Inception | Temporal Mixup | $\mathcal{L}_C + \mathcal{L}_{\text{Contrastive}} + H$ |
| Raincoat | 1D CNN | Frequency encoder, Decoder | $\mathcal{L}_C + \mathcal{L}_R + \mathcal{L}_{\text{Sinkhorn}}$ |
| InceptionRain | Inception | Frequency encoder, Decoder | $\mathcal{L}_C + \mathcal{L}_R + \mathcal{L}_{\text{Sinkhorn}}$ |
| InceptionTime | Inception | - | $\mathcal{L}_C$ |
| OTDA | - | Transport map | - |

state-of-the-art approach for time series classification is trained with the source domain data, and the target domain data is only used for hyperparameter tuning. We also use it in this benchmark as a backbone for other domain adaptation algorithms, referred to as the "Inception" backbone, allowing the comparison of different UDA algorithms independently of their backbones.

**Adversarial domain adaptation**   This technique uses a discriminator to train the backbone, enforcing domain adaptation. During training the discriminator learns how to distinguish between source domain and target domain samples. Simultaneously, the backbone learns how to fool the discriminator. This technique was inspired by Generative Adversarial Networks (Goodfellow et al. (2014)) and originally proposed for domain adaptation in Ganin et al. (2016).

Variational Recurrent Adversarial Deep Domain Adaptation (VRADA) from Purushotham et al. (2016), Convolutional deep Domain Adaptation model for Time Series data (CoDATS) from Wilson et al. (2020) and InceptionDANN (proposed in this work) all use adversarial domain adaptation. For all these algorithms, the architecture of the classifier and discriminator consists of a stack of fully connected layers. The main difference between these algorithms lies in their backbones.

VRADA is a domain adaptation algorithm tailored for time series which enforces domain adaptation through adversarial learning. The backbone of VRADA is a Variational Recurrent Neural Network (VRNN) (Chung et al. (2015)), which integrates elements of a Variational Auto-Encoder (Kingma & Welling (2014)) into a Recurrent Neural Network in order to achieve the variability observed in highly structured time series data. CoDATS proposes the use of a 1-dimensional fully convolutional neural network (1D CNN) to build the backbone. This architecture significantly reduces training time and offers better results in comparison to that of VRADA. InceptionDANN is proposed to compare adversarial domain adaptation to other algorithms with the same backbone. The algorithm is the same as CoDATS, but uses the Inception backbone. Additionally, we propose InceptionC-DAN, a deep UDA method based on Conditional DANN (Long et al. (2018)), where multilinear conditioning is used to capture the cross-covariance between feature representations and classifier predictions. This method also uses the Inception backbone.

**Contrastive learning**   Contrastive learning aims at aligning the predictions made by the model for pairs of samples coming from two different domains, thus enforcing domain adaptation. CoT-Mix (Eldele et al., 2023) implements contrastive learning between the soft probabilities of source domain samples and their source dominant counterparts and between the soft probabilities of target domain samples and their target dominant counterparts. Source dominant and target dominant samples are produced through the *temporal mixup* operation.

The temporal mixup operation selects a source (or target) domain sample and a target (respectively source) domain sample. The temporal mixup is produced by a component-wise weighted sum between the first sample and a moving average of the second sample. Each temporal mixup should preserve the characteristics of the *dominant* domain while considering the temporal information from the other, *less dominant*, domain.

CoTMix's bakcbone is a 1D CNN. InceptionMix, proposed in this work, applies the same learning algorithm as CoTMix while employing the Inception backbone.

**Frequency domain analysis**  Raincoat is a novel method for UDA in time series proposed by He et al. (2023) in which the temporal features and frequency features of the time series are analyzed separately. Raincoat consists of three different modules: an encoder, a decoder and a classifier.

The main novelty is in the encoder, which treats separately time and frequency features. Time features are analyzed by a traditional backbone, such as a 1D CNN, while the frequency features are extracted by: (1) smoothing, (2) applying a discrete Fourier transform, (3) multiplying by a learned weight matrix, (4) transforming into amplitude and phase components, and (5) concatenating. Once the frequency and time features are extracted, these are concatenated and passed on to a classifier which is trained for the main classification task.

The role of the decoder is to reconstruct the input samples from the features extracted by the encoder. A reconstruction loss computed over both source and target domain samples forces the encoder to produce an accurate representation of the target samples. Additionally, the Sinkhorn divergence between the extracted source domain and target domain features is computed and used to align the target and source domain representations. InceptionRain, proposed in this work, uses the Inception backbone as a time feature encoder for the Raincoat algorithm.

**Non-deep UDA**  In addition to the algorithms mentioned above, we consider Optimal Transport Domain Adaptation (OTDA) (Courty et al., 2015) as a non-deep approach, which consists in aligning the source distribution over the target one using the Optimal Transport plan. Note however that this approach is not originally designed for time series.

### 2.3  TUNING MODELS WITHOUT LABELS

The hyperparameter tuning step is particularly challenging in UDA, given the absence of labels in the target domain for model evaluation, and can drastically impact the methods' performances (Musgrave et al., 2022). While it is a common practice in many UDA papers to manually set hyperparameters for each scenario (Tzeng et al., 2017; Wilson et al., 2020), three standard approaches for an automatic selection of hyperparameters are presented below.

**Target Risk**  Several methods (Long et al., 2015; Courty et al., 2015; Saito et al., 2018) rely on target risk to select models, where the risk is computed on target labels, which are assumed to be available at least partially. However, this method is questionable because it will not be applicable for real UDA experiments and serves more as an oracle and an upper bound on performance.

**Source Risk**  Ganin & Lempitsky (2015) selects hyperparameters by using the empirical source risk. Although it should lead in theory to a misestimation of the target risk in the presence of a large domain gap, Musgrave et al. (2022) have shown that it results in good performance on UDA for computer vision.

**Importance Weighted Cross Validation (IWCV)**  A more theoretically sound approach to estimate the target risk is proposed by Sugiyama et al. (2007) and applied in Eldele et al. (2023); Long et al. (2018), where each source sample $\mathbf{X}$ is weighted by the ratio between the probability density of the target $p_T(\mathbf{X})$ and the source $p_S(\mathbf{X})$. The authors prove that the following equality holds under the covariant shift assumption with any loss function $\mathcal{L}$ and classifier $f$

$$\text{Target loss}(f, \mathcal{L}) = \mathbb{E}_{(\mathbf{X},\mathbf{y}) \sim p_S} \left[ \frac{p_T(\mathbf{X})}{p_S(\mathbf{X})} \mathcal{L}(\mathbf{y}, f(\mathbf{X})) \right].$$

Taking $\mathcal{L}$ as the $0-1$ loss function leads to a proxy of the target risk. In practice, the performance of IWCV are strongly limited by the unverifiability of covariate shift assumption and the difficulty of estimating the densities.

Although other methods exist in the literature, such as Zhong et al. (2010); You et al. (2019); Robbiano et al. (2022); Chuang et al. (2020); Saito et al. (2021); Musgrave et al. (2022), we choose the approaches that offer the best practicality, robustness, lowest computational cost, and are well studied in the literature.

## 3 DATASETS USED FOR EVALUATION

The most commonly used datasets in UDA for TSC revolve around human activity recognition, fault detection and sleep stage prediction: (1) Human Activity Recognition (HAR); (2) Heterogeneity Human Activity Recognition (HHAR); (3) Machine Fault Diagnosis (MFD); (4) Sleep Stage; (5) Wireless Sensor Data Mining (WISDM) (Anguita et al., 2013; Stisen et al., 2015; Lessmeier et al., 2016; Ragab et al., 2023; Kwapisz et al., 2011). In an effort to increase the diversity in terms of applications as well as having a larger sample for comparing the classifiers, we introduce seven new datasets. Table 2, presents an overview of each dataset, including important statistics like the number of domains, the number of classes, the length of each time series, the number of channels and the four different themes: machinery, motion, medical and remote sensing. The following is a brief summary describing each of the *new* datasets proposed in this paper with a focus on: (1) the time series classification task and (2) the domain adaptation problem. For datasets where the number of possible UDA scenarios exceeds five, we have taken a random selection of five scenarios to limit the number of experiments which increases exponentially with each additional scenario.

Table 2: Description of unsupervised domain adaptation time series datasets. Bold indicates the new datasets proposed in this benchmark.

| Dataset | Domains | Classes | Length | Channels | Theme |
|---|---|---|---|---|---|
| **ford** | 2 | 2 | 500 | 1 | machinery |
| **cwrBearing** | 4 | 4 | 512 | 1 | machinery |
| mfd | 4 | 3 | 5120 | 1 | machinery |
| **ptbXLecg** | 3 | 5 | 1000 | 12 | medical |
| **ultrasoundMuscleContraction** | 8 | 2 | 3000 | 1 | medical |
| sleepStage | 20 | 5 | 3000 | 1 | medical |
| **OnHWeq** | 2 | 15 | 64 | 13 | motion |
| **sportsActivities** | 8 | 19 | 125 | 45 | motion |
| hhar | 9 | 6 | 128 | 3 | motion |
| wisdm | 36 | 6 | 128 | 6 | motion |
| har | 30 | 6 | 128 | 9 | motion |
| **miniTimeMatch** | 4 | 8 | 39 | 10 | remote sensing |

**Ford** (Dau et al., 2019) was initially employed in the IEEE World Congress on Computational Intelligence in 2008 as part of a competition. The primary classification task involved diagnosing specific symptoms within an automotive subsystem. Each instance in the dataset comprises 500 measurements of engine noise alongside a corresponding classification label of whether or not an anomaly exists in the engine. There are two distinct domains: (1) FordA where data were gathered under typical operational conditions, characterized by minimal noise interference; (2) FordB where data were collected under noisy conditions. **CWR Bearing** (Zhang et al., 2019) consists of motor vibration data collected using accelerometers placed at the 12 o'clock position at both the drive end and fan end of the motor housing in order to detect normal and faulty bearings, with one normal class and 3 fault classes. The data was collected at 12,000 and 48,000 samples per second for drive end bearing experiments. Five distinct scenarios are generated from different motor conditions. **PTB XL ECG** (Wagner et al., 2020) is a collection of 549 high-resolution 15 channels ECGs. The data was recorded at different clinical sites and gathers 294 subjects, including healthy subjects as well as patients with a variety of heart diseases representing the 5 different classes. Five domain adaptation scenarios are created from 4 clinical sites. **Ultrasound Muscle Contraction** (Brausch et al., 2022) is a collection of 21 sets of one-dimensional ultrasound raw radio frequency data (A-Scans) measured on calf muscles of 8 healthy volunteers, each A-Scan consisting of 3000 amplitude values. The purpose is to determine whether the muscle is contracted or not. Each individual subject is considered as an independent domain. **Online Handwritten Equations** (Ott et al., 2022) consists of recognizing handwritten equations based on multivariate time series data captured from sensors placed on a sensor enhanced pen. With 12 possible equations and 55 different writers, the time series classification task is based on 12 labels while domain adaptation scenarios are split based on the writer's identity. **Sport Activities** (Altun et al., 2010) comprises motion sensor data of 19 daily sports activities performed by 8 subjects in their own style for 5 minutes. Data was recorded using five Xsens MTx sensor units placed on the torso, arms, and legs. Each individual subject is considered as an independent domain. **Mini Time Match** (Nyborg et al., 2022) is a crop-type

mapping that covers 4 regions across Europe. It comprises a series of time-stamped multi-spectral measurements derived from satellite imagery captured at specific geographic coordinates. The aim is to recognize the type of crop of a parcel among 8 categories, such as corn or wheat, each region being an independent domain.

## 4 EXPERIMENTAL SETUP

The framework that we have developed consists of five stages, similar to a traditional machine learning pipeline: (1) **Loading**: raw time series datasets are downloaded from their original source that we separate into train, validation and test sets if not already split. In the source domain, labels were utilized to ensure an equitable distribution of classes across all three sets. This stratification was not performed in the target domain, where labels are not supposed to be available. However, in line with supervised ML conventions, the class proportions between the test set and the training/validation sets remain consistent in the target domain. Additionally, when possible, temporal causality is ensured for all the splits. (2) **Preprocessing**: each raw time series is preprocessed in the same way for all methods and following the recommendations of the paper that proposed the dataset (e.g. z-normalization). (3) **Tuning**: for each couple of (dataset, classifier), we search for the best hyperparameters using the three methods presented in Section 2.3. For IWCV, the marginal distributions $p_T(\mathbf{X})$ and $p_S(\mathbf{X})$ are estimated by a 5-Gaussian mixture with $\mathcal{L}$ denoting the cross-entropy. For all 3 methods, the source or target risk is estimated using the validation set, which is unseen during training. (4) **Training**: for each of the three model selection methods, we take the best hyperparameter set, we re-train the model, check-pointing the weights at each epoch using the same metric and validation set used during tuning. (5) **Evaluation**: for each trained model, we evaluate the final metric over the previously unseen test set (source and target). Thus, contrary to some DA settings, the test data are never seen during training nor validation.

To ensure fairness across all algorithms, we fixed the hyperparameter tuning budget for all experiments to 12 hours of GPU time. Similarly, during the training stage, we specify the maximum training budget to 2 hours of GPU time for each model on a given dataset for a given set of hyperparameters. Our benchmark results in $1458$ different experiments, with 12 datasets (54 domain adaptation scenarios), 3 hyperparameter tuning methods and 9 deep UDA algorithms. Thus, the total sequential runtime of this benchmark is approximately $8748$ hours, corresponding to almost one year. Finally, in the spirit of reproducible machine learning research, we are planning to open source the code upon acceptance of the paper.

## 5 RESULTS AND ANALYSIS OF THE BENCHMARK

In the following subsections we start by presenting an overview of the results while comparing multiple classifiers over multiple datasets, as well as analyzing the best classifier's accuracy. In later subsections, we perform a statistical analysis between the various hyperparameter tuning methods used for each experiment and we showcase the impact of the neural network architecture (backbone) when using the same adaptation technique. The complete results for each dataset are provided in Appendix A.1. Note that some of our results differ in accuracy from the original papers, which presented the following limitations: (1) hard-coded hyperparameters were set for a given scenario (He et al., 2023); or (2) target labels were used for tuning the hyperparameters (Wilson et al., 2020); or (3) the time series segmentation did not respect temporal causality thus introducing data leakage (Eldele et al., 2023).

### 5.1 COMPARISON OF CLASSIFIERS OVER ALL DATASETS

For the overall comparison we carry out the analysis depending on the choice of the hyperparameter tuning method. Following the classical approach for comparing multiple time series classifiers (Bagnall et al., 2017), we first perform a Friedman test over all the algorithms which rejects the null hypothesis over the different repeated measures (Friedman, 1940). Each repeated measure corresponds to an unsupervised domain adaptation scenario. For comparing multiple classifiers over multiple datasets, we make use of the critical difference diagram (Demšar, 2006). However we omit the post-hoc analysis based on Wilcoxon-signed rank test with Holm's alpha correction as this introduces artifacts into the diagram (Lines et al., 2018).

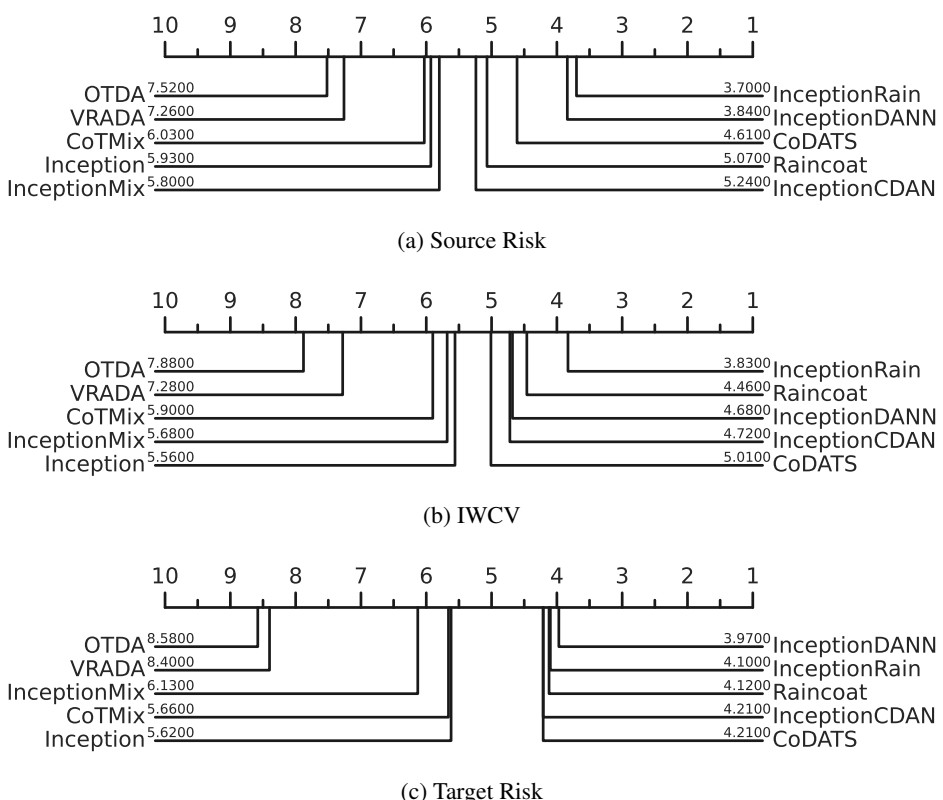

Figure 1: Average rank diagrams based on accuracy for different hyperparameter tuning methods

Figure 1a, showcases the average rank of all classifiers when using the Source Risk to tune the hyperparameters of the algorithms. We can identify InceptionRain with the highest average rank among all 9 models. In addition to these 9 classifiers, we have moved the results for a single model called SASA (Cai et al., 2021) (with both LSTM and Inception backbones) into Appendix A.4 where we provide detailed explanation for these additional results. Furthermore in Figure 1b, when tuning the hyperparameters using the IWCV method, InceptionRain still has the best performance in terms of accuracy over all the datasets on average. In Figure 1c, we notice a small drop in InceptionRain's performance relative to the other classifiers when using the Target Risk for selecting the best hyperparameters. Moreover, we notice across the three diagrams, three cluster of algorithms starting with VRADA and OTDA comprising the worst two classifiers. The second cluster consists of InceptionMix, CoTMix and Inception showing very similar average ranking. This can be explained by the fact that InceptionMix and CoTMix differ solely with the neural network backbone architecture using the same contrastive loss approach presented by CoTMix in Eldele et al. (2023). Surprisingly, Inception was able to achieve similar results to CoTMix and InceptionMix without employing a domain adaptation technique, suggesting that the method proposed by CoTMix is not able to generalize to the larger benchmark presented here, compared to the results provided in CoTMix' original paper. Additionally, we can pinpoint a third cluster of algorithms based on adversarial training as well as the Raincoat approach. Nevertheless InceptionRain remains a constant top performing algorithm across the three different hyperparameter tuning methods. Finally, we can notice across the three diagrams that the intra-cluster distance (in terms of average rank) decreases significantly when using the Target Risk for hyperparameter tuning. The latter observation suggests that a good hyperparameter tuning method can close the gap between several classifiers, which is in line with the recent domain adaptation reviews for computer vision (Musgrave et al., 2021). Other average rank diagrams, using the F1-score, are depicted in Appendix A.2 with similar conclusion.

Following the overall comparison over all the datasets, we provide a pairwise accuracy plot comparison of InceptionRain (best classifier) against Inception and Raincoat. Figure 2a showcases the

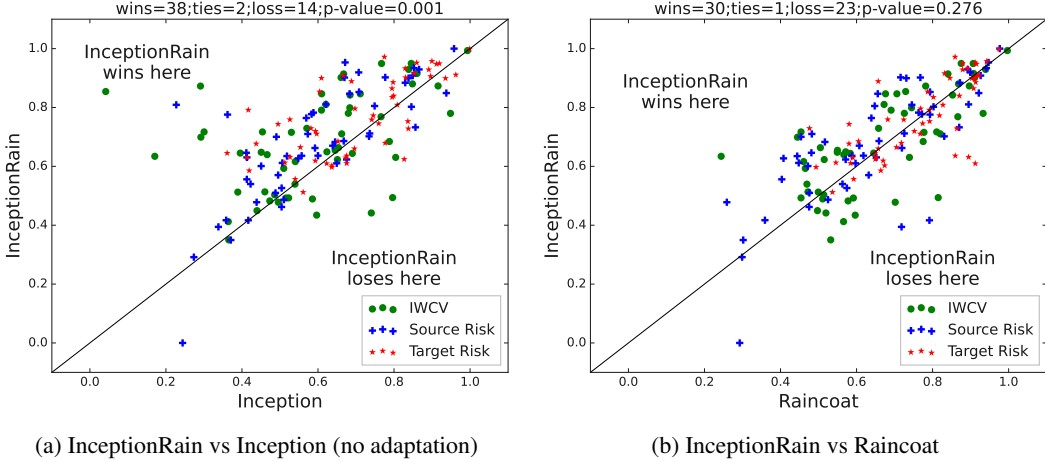

Figure 2: Pairwise accuracy comparison of InceptionRain against Inception and Raincoat. The statistical analysis and comparison above the figures are computed using only results based on IWCV.

impact of the UDA method proposed in Raincoat (He et al., 2023), by comparing Inception's backbone without DA against InceptionRain. The latter model achieves significantly better accuracy than its pure backbone counterpart with a Win/Tie/Loss equal to 38/2/14 and a $p-value = 0.001$, demonstrating the advantage of using domain adaptation. Finally, Figure 2b presents the difference between Raincoat and InceptionRain with a Win/Tie/Loss equal to 30/1/23 and a $p-value = 0.276$ suggesting that with a change in backbone one is not able to reject the null hypothesis, thus providing evidence that the domain adaption technique, rather than its backbone, is the main reason behind the top performance of InceptionRain in this benchmark.

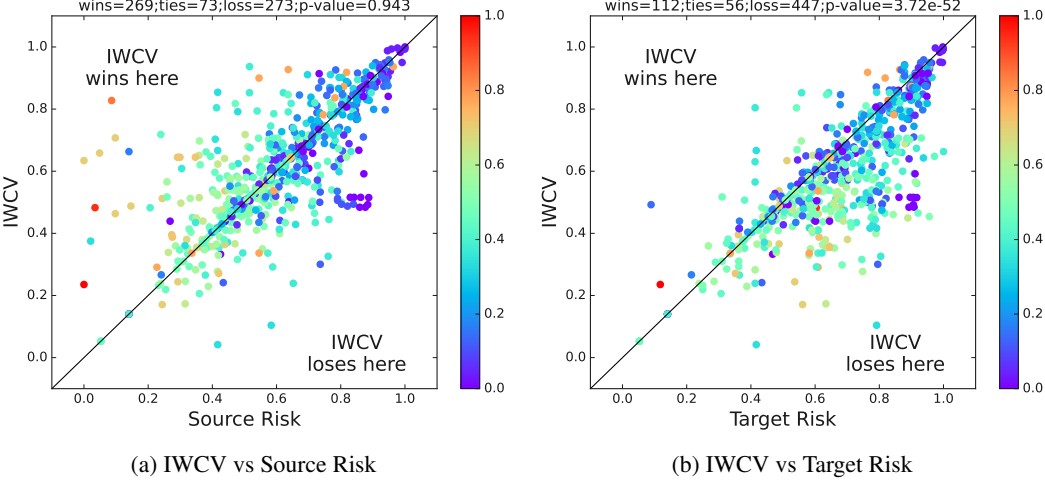

Figure 3: Pairwise accuracy comparison of three hyperparameter tuning methods. Colors acts as a proxy for the shift between source and target domains. They are generated based on Inception's relative accuracy between source and target. Purple meaning that Inception performs equally in source and target, suggesting a high similarity between the two domains.

## 5.2 COMPARISON OF HYPERPARAMETER TUNING METHODS

We now provide some insight about the choice of approach for tuning the hyperparameters. Figure 3 displays the pairwise accuracy comparison with IWCV against Source Risk and Target Risk. In addition, we have included a colormap based on the variations in Inception's accuracy between the source and the target test set as a surrogate for estimating the degree of shift between source and

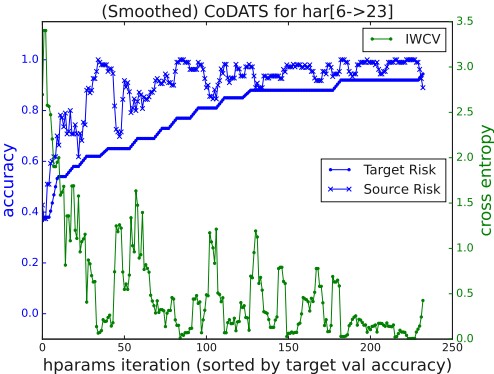 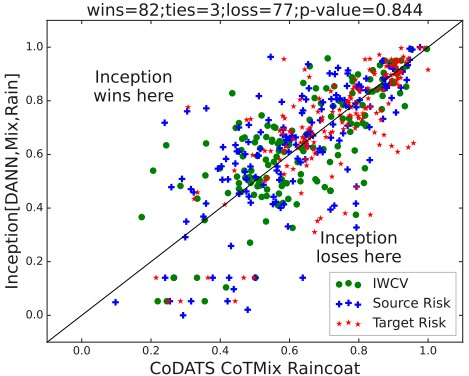

Figure 4: Accuracy and cross entropy loss during the hyperparameter tuning stage

Figure 5: Pairwise comparison of Inception backbone vs other backbones

target for each scenario, with red hinting at a large shift and blue at a small shift. Figure 3a indicates that IWCV and Source Risk are not significantly different with a $p - value$ close to 1, but the colors indicate that the larger the shift between source and target, the larger the difference between IWCV and Source Risk. This suggests that IWCV is more beneficial whenever the shift between source and target is large, a shift that should be theoretically corrected by IWCV. However, Figure 3b shows that IWCV is still significantly worse than Target Risk (which leverages the labels), thus suggesting that there is still room for improving the hyperparameter tuning methods and further research into this direction seems promising. In addition to the latter overall comparison, Figure 4 displays one example of hyperparameter tuning where we can see that there exists a correlation between the target risk and the two proxy used. More examples similar to Figure 4 are given in Appendix A.2.5, further validating the current trend seen for CoDATS over the HAR dataset.

## 5.3 COMPARISON OF BACKBONES

This final subsection takes a deeper look into the different backbones, investigating how impactful the choice of the backbone is to the domain adaptation problem. We saw in Figure 2b that for the Raincoat approach, changing the backbone to Inception (with InceptionRain) does not have a significant impact on target accuracy. This conclusion holds for CoTMix and CoDATS as shown in Appendix A.2.6. To further validate this across all backbones, Figure 5 displays the pairwise accuracy comparison between CoDATS, Raincoat and CoTMix against the three same methods based on the Inception backbone. The statistical comparison indicates a small difference with a $p - value$ above 0.8 thus suggesting that given the current benchmark, backbones do not have a significant impact and the main difference stems from the UDA technique itself.

## 6 CONCLUSION

This study presents a comprehensive benchmark evaluation of contemporary algorithms for deep unsupervised domain adaptation in the context of time series classification. Additionally, we introduce novel datasets designed to establish a strong baseline for performance evaluation, facilitating a clearer understanding of the research landscape. This benchmark enables a fair comparison among algorithms by employing various hyperparameter tuning methods maintaining a consistent time budget for the tuning process.

In the future, we aspire to expand the framework's capabilities to accommodate a wider array of hyperparameter tuning techniques. Indeed our findings showcased the significance of careful model selection methods, thereby encouraging further exploration and refinement of hyperparameter tuning strategies within unsupervised domain adaptation for time series data. Finally, we aspire to address another outstanding issue related to the interplay between the degree of shift and the performance of UDA approaches. This challenge centers on accurately estimating the shift, a task that is inherently difficult overall when dealing specifically with multivariate time series data.

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

# A APPENDIX

## A.1 COMPLETE RESULTS PER DATASET

### A.1.1 ALL DATASETS PERFORMANCES

The different datasets and scenario are evaluated in more details in this section. Results of the accuracy per dataset per method of hyperparameter tuning are presented in 36 different tables from 3 to 38. The source only model is trained using Inception (Ismail Fawaz et al., 2020) on the source data without consideration of any adaptation. It is then tested on target-domain data. This could be seen as a lower bound on the different UDA algorithms. The target only model is trained using using Inception (Ismail Fawaz et al., 2020) on the target domain with class labels revealed. This can be seen as an upper bound on the different UDA algorithms. As explained in the experiments setup (section 4), due to the runtime constraint, we report the model's accuracy after only one run of the best selected hyperprameter (per method: source risk, target risk and the IWCV).

It is also worth noting that the results sometimes present large discrepancies in performance for different source-target couples within a dataset. Such discrepancies come mostly from the differences in domains. For instance, FordA is noiseless data while FordB is noisy, thus performance tend to be better for FordB → FordA than for FordA → FordB. Other examples of differences between domains are visible for HHAR dataset, which including elder people who present more noisy data, and on miniTimeMatch dataset, where two crops come from France (FR1 and FR2), having a similar weather, while the other crops come from Denmark (DK1) and Austria (AT1).

Table 3: Accuracy results on cwrBearing dataset using IWCV for the model selection and evaluated on Target Test

|  | $0 \rightarrow 1$ | $3 \rightarrow 0$ | $0 \rightarrow 3$ | $1 \rightarrow 3$ | $2 \rightarrow 0$ | Avg |
|---|---|---|---|---|---|---|
| Source only | 0.848 | 0.767 | 0.788 | 0.948 | 0.916 | $0.853 \pm 0.07$ |
| CoDATS | 0.798 | 0.890 | **0.830** | 0.872 | 0.710 | $0.82 \pm 0.064$ |
| CoTMix | 0.861 | 0.871 | 0.782 | 0.640 | 0.861 | $0.803 \pm 0.088$ |
| InceptionCDAN | 0.816 | 0.926 | 0.824 | 0.893 | 0.853 | $0.862 \pm 0.042$ |
| InceptionDANN | 0.898 | 0.942 | **0.834** | 0.904 | **0.958** | **0.907 ± 0.043** |
| InceptionMix | **0.933** | 0.844 | 0.828 | 0.867 | 0.850 | $0.864 \pm 0.036$ |
| InceptionRain | 0.880 | **0.949** | 0.684 | 0.780 | 0.873 | $0.833 \pm 0.092$ |
| OTDA | 0.535 | 0.709 | 0.603 | 0.716 | 0.685 | $0.65 \pm 0.07$ |
| Raincoat | 0.868 | 0.907 | 0.775 | **0.933** | 0.897 | $0.876 \pm 0.055$ |
| VRADA | 0.475 | 0.492 | 0.430 | 0.428 | 0.505 | $0.466 \pm 0.032$ |
| Target only | 0.991 | 0.991 | 0.988 | 0.991 | 0.995 | $0.991 \pm 0.002$ |

Table 4: Accuracy results on cwrBearing dataset using Source Risk for the model selection and evaluated on Target Test

|  | $0 \rightarrow 1$ | $3 \rightarrow 0$ | $0 \rightarrow 3$ | $1 \rightarrow 3$ | $2 \rightarrow 0$ | Avg |
|---|---|---|---|---|---|---|
| Source only | 0.838 | 0.849 | 0.734 | 0.938 | 0.852 | $0.842 \pm 0.065$ |
| CoDATS | **0.938** | 0.740 | **0.881** | 0.868 | 0.871 | $0.86 \pm 0.065$ |
| CoTMix | 0.888 | 0.907 | 0.806 | 0.800 | 0.801 | $0.84 \pm 0.047$ |
| InceptionCDAN | 0.849 | 0.784 | 0.819 | 0.938 | 0.891 | $0.856 \pm 0.054$ |
| InceptionDANN | 0.878 | **0.923** | 0.848 | **0.947** | **0.950** | **0.909 ± 0.04** |
| InceptionMix | 0.929 | 0.757 | 0.853 | 0.860 | 0.847 | $0.849 \pm 0.055$ |
| InceptionRain | 0.900 | 0.908 | 0.702 | 0.849 | 0.934 | $0.859 \pm 0.083$ |
| OTDA | 0.789 | 0.709 | 0.689 | 0.809 | 0.597 | $0.719 \pm 0.076$ |
| Raincoat | 0.731 | **0.926** | 0.831 | 0.922 | 0.941 | $0.87 \pm 0.08$ |
| VRADA | 0.465 | 0.498 | 0.430 | 0.477 | 0.467 | $0.467 \pm 0.022$ |
| Target only | 0.991 | 0.991 | 0.988 | 0.991 | 0.995 | $0.991 \pm 0.002$ |

Table 5: Accuracy results on cwrBearing dataset using Target Risk for the model selection and evaluated on Target Test

|  | $0 \to 1$ | $3 \to 0$ | $0 \to 3$ | $1 \to 3$ | $2 \to 0$ | Avg |
|---|---|---|---|---|---|---|
| Source only | 0.935 | 0.891 | 0.799 | 0.963 | 0.912 | $0.9 \pm 0.056$ |
| CoDATS | 0.935 | 0.910 | 0.854 | 0.895 | 0.907 | $0.9 \pm 0.027$ |
| CoTMix | 0.909 | 0.791 | 0.801 | 0.893 | 0.844 | $0.848 \pm 0.047$ |
| InceptionCDAN | 0.922 | 0.868 | 0.847 | 0.936 | 0.898 | $0.894 \pm 0.033$ |
| InceptionDANN | **0.942** | 0.879 | 0.846 | **0.954** | 0.938 | **0.912 ± 0.042** |
| InceptionMix | 0.926 | 0.879 | 0.845 | 0.896 | 0.907 | $0.891 \pm 0.027$ |
| InceptionRain | 0.899 | **0.929** | 0.840 | 0.911 | **0.948** | $0.905 \pm 0.037$ |
| OTDA | 0.790 | 0.733 | 0.745 | 0.825 | 0.695 | $0.758 \pm 0.045$ |
| Raincoat | 0.903 | 0.879 | **0.870** | 0.922 | 0.915 | $0.898 \pm 0.02$ |
| VRADA | 0.466 | 0.089 | 0.430 | 0.501 | 0.550 | $0.407 \pm 0.164$ |
| Target only | 0.991 | 0.991 | 0.988 | 0.991 | 0.995 | $0.991 \pm 0.002$ |

Table 6: Accuracy results on ford dataset using IWCV for the model selection and evaluated on Target Test

|  | $FordB \to FordA$ | $FordA \to FordB$ | Avg |
|---|---|---|---|
| Source only | 0.516 | 0.796 | $0.656 \pm 0.14$ |
| CoDATS | 0.484 | 0.488 | $0.486 \pm 0.002$ |
| CoTMix | 0.515 | 0.512 | $0.514 \pm 0.002$ |
| InceptionCDAN | 0.516 | 0.500 | $0.508 \pm 0.008$ |
| InceptionDANN | 0.516 | 0.494 | $0.505 \pm 0.011$ |
| InceptionMix | 0.484 | 0.506 | $0.495 \pm 0.011$ |
| InceptionRain | 0.492 | 0.494 | $0.493 \pm 0.001$ |
| OTDA | 0.484 | 0.494 | $0.489 \pm 0.005$ |
| Raincoat | **0.592** | **0.815** | **0.704 ± 0.111** |
| VRADA | 0.492 | 0.506 | $0.499 \pm 0.007$ |
| Target only | 0.953 | 0.827 | $0.89 \pm 0.063$ |

Table 7: Accuracy results on ford dataset using Source Risk for the model selection and evaluated on Target Test

|  | $FordB \to FordA$ | $FordA \to FordB$ | Avg |
|---|---|---|---|
| Source only | 0.830 | 0.846 | $0.838 \pm 0.008$ |
| CoDATS | 0.877 | 0.821 | $0.849 \pm 0.028$ |
| CoTMix | **0.887** | 0.790 | $0.838 \pm 0.048$ |
| InceptionCDAN | 0.842 | 0.735 | $0.788 \pm 0.053$ |
| InceptionDANN | 0.862 | **0.840** | **0.851 ± 0.011** |
| InceptionMix | 0.859 | 0.790 | $0.824 \pm 0.034$ |
| InceptionRain | **0.884** | 0.802 | $0.843 \pm 0.041$ |
| OTDA | 0.492 | 0.475 | $0.484 \pm 0.009$ |
| Raincoat | 0.872 | 0.802 | $0.837 \pm 0.035$ |
| VRADA | 0.491 | 0.500 | $0.496 \pm 0.005$ |
| Target only | 0.953 | 0.827 | $0.89 \pm 0.063$ |

Table 8: Accuracy results on ford dataset using Target Risk for the model selection and evaluated on Target Test

|  | $FordB \rightarrow FordA$ | $FordA \rightarrow FordB$ | Avg |
|---|---|---|---|
| Source only | 0.896 | 0.809 | $0.852 \pm 0.043$ |
| CoDATS | 0.892 | 0.778 | $0.835 \pm 0.057$ |
| CoTMix | **0.917** | **0.821** | $\mathbf{0.869 \pm 0.048}$ |
| InceptionCDAN | 0.906 | 0.759 | $0.832 \pm 0.074$ |
| InceptionDANN | 0.861 | 0.809 | $0.835 \pm 0.026$ |
| InceptionMix | 0.908 | 0.772 | $0.84 \pm 0.068$ |
| InceptionRain | 0.906 | 0.809 | $0.858 \pm 0.048$ |
| OTDA | 0.495 | 0.562 | $0.528 \pm 0.034$ |
| Raincoat | **0.911** | **0.827** | $\mathbf{0.869 \pm 0.042}$ |
| VRADA | 0.516 | 0.500 | $0.508 \pm 0.008$ |
| Target only | 0.953 | 0.827 | $0.89 \pm 0.063$ |

Table 9: Accuracy results on har dataset using IWCV for the model selection and evaluated on Target Test

|  | $7 \rightarrow 13$ | $12 \rightarrow 16$ | $9 \rightarrow 18$ | $2 \rightarrow 11$ | $6 \rightarrow 23$ | Avg |
|---|---|---|---|---|---|---|
| Source only | 0.838 | 0.609 | 0.291 | 0.684 | 0.661 | $0.617 \pm 0.18$ |
| CoDATS | 0.919 | 0.600 | 0.927 | 0.663 | 0.902 | $0.802 \pm 0.141$ |
| CoTMix | 0.869 | **0.818** | 0.900 | **0.937** | 0.911 | $\mathbf{0.887 \pm 0.041}$ |
| InceptionCDAN | 0.889 | 0.618 | 0.836 | 0.853 | 0.643 | $0.768 \pm 0.114$ |
| InceptionDANN | 0.828 | 0.655 | 0.918 | 0.674 | 0.866 | $0.788 \pm 0.105$ |
| InceptionMix | 0.818 | 0.745 | **0.936** | 0.853 | 0.902 | $0.851 \pm 0.067$ |
| InceptionRain | 0.929 | 0.791 | 0.873 | 0.842 | 0.902 | $0.867 \pm 0.048$ |
| OTDA | 0.475 | 0.336 | 0.336 | 0.895 | 0.598 | $0.528 \pm 0.208$ |
| Raincoat | **0.939** | 0.691 | 0.782 | 0.853 | **0.911** | $0.835 \pm 0.09$ |
| VRADA | 0.727 | 0.409 | 0.536 | 0.337 | 0.688 | $0.539 \pm 0.152$ |
| Target only | 1.000 | 0.982 | 1.000 | 1.000 | 1.000 | $0.996 \pm 0.007$ |

Table 10: Accuracy results on har dataset using Source Risk for the model selection and evaluated on Target Test

|  | $7 \rightarrow 13$ | $12 \rightarrow 16$ | $9 \rightarrow 18$ | $2 \rightarrow 11$ | $6 \rightarrow 23$ | Avg |
|---|---|---|---|---|---|---|
| Source only | 0.707 | 0.600 | 0.227 | 0.621 | 0.670 | $0.565 \pm 0.173$ |
| CoDATS | 0.879 | 0.591 | 0.636 | 0.832 | 0.804 | $0.748 \pm 0.114$ |
| CoTMix | 0.768 | 0.645 | 0.545 | 0.516 | 0.696 | $0.634 \pm 0.094$ |
| InceptionCDAN | 0.818 | **0.727** | 0.782 | 0.516 | 0.670 | $0.703 \pm 0.106$ |
| InceptionDANN | 0.889 | 0.655 | 0.818 | 0.579 | 0.839 | $0.756 \pm 0.118$ |
| InceptionMix | 0.707 | 0.582 | **0.964** | 0.621 | 0.795 | $0.734 \pm 0.137$ |
| InceptionRain | **0.919** | 0.636 | 0.809 | 0.811 | **0.902** | $\mathbf{0.815 \pm 0.1}$ |
| OTDA | 0.535 | 0.573 | 0.545 | 0.853 | 0.795 | $0.66 \pm 0.136$ |
| Raincoat | 0.899 | 0.618 | 0.745 | **0.895** | 0.768 | $0.785 \pm 0.105$ |
| VRADA | 0.616 | 0.409 | 0.591 | 0.495 | 0.580 | $0.538 \pm 0.076$ |
| Target only | 1.000 | 0.982 | 1.000 | 1.000 | 1.000 | $0.996 \pm 0.007$ |

Table 11: Accuracy results on har dataset using Target Risk for the model selection and evaluated on Target Test

| | $7 \rightarrow 13$ | $12 \rightarrow 16$ | $9 \rightarrow 18$ | $2 \rightarrow 11$ | $6 \rightarrow 23$ | Avg |
|---|---|---|---|---|---|---|
| Source only | 0.919 | 0.745 | 0.609 | 0.853 | 0.830 | $0.791 \pm 0.107$ |
| CoDATS | 0.879 | **0.873** | 0.864 | **1.000** | 0.875 | **0.898** $\pm$ **0.051** |
| CoTMix | 0.919 | 0.845 | 0.818 | 0.958 | 0.929 | **0.894** $\pm$ **0.053** |
| InceptionCDAN | 0.909 | 0.818 | 0.891 | 0.558 | 0.938 | $0.823 \pm 0.138$ |
| InceptionDANN | 0.899 | 0.673 | 0.764 | 0.916 | 0.893 | $0.829 \pm 0.095$ |
| InceptionMix | **0.949** | 0.682 | **0.955** | 0.642 | **0.964** | $0.838 \pm 0.145$ |
| InceptionRain | **0.949** | 0.773 | 0.900 | 0.916 | 0.902 | $0.888 \pm 0.06$ |
| OTDA | 0.808 | 0.609 | 0.582 | 0.874 | 0.607 | $0.696 \pm 0.121$ |
| Raincoat | 0.879 | 0.818 | 0.845 | 0.832 | 0.911 | $0.857 \pm 0.034$ |
| VRADA | 0.727 | 0.609 | 0.609 | 0.621 | 0.661 | $0.645 \pm 0.045$ |
| Target only | 1.000 | 0.982 | 1.000 | 1.000 | 1.000 | $0.996 \pm 0.007$ |

Table 12: Accuracy results on hhar dataset using IWCV for the model selection and evaluated on Target Test

| | $e \rightarrow f$ | $g \rightarrow h$ | $c \rightarrow d$ | $i \rightarrow a$ | $a \rightarrow b$ | Avg |
|---|---|---|---|---|---|---|
| Source only | 0.766 | 0.510 | 0.585 | 0.473 | 0.540 | $0.575 \pm 0.102$ |
| CoDATS | **0.869** | **0.818** | 0.717 | 0.680 | 0.668 | **0.75** $\pm$ **0.079** |
| CoTMix | 0.840 | 0.538 | **0.740** | 0.559 | **0.698** | $0.675 \pm 0.113$ |
| InceptionCDAN | 0.690 | 0.421 | 0.474 | 0.576 | 0.585 | $0.549 \pm 0.094$ |
| InceptionDANN | 0.854 | 0.548 | 0.572 | 0.500 | 0.626 | $0.62 \pm 0.124$ |
| InceptionMix | 0.835 | 0.561 | 0.598 | **0.700** | 0.620 | $0.663 \pm 0.097$ |
| InceptionRain | 0.769 | 0.593 | 0.489 | 0.483 | 0.616 | $0.59 \pm 0.104$ |
| OTDA | 0.596 | 0.411 | 0.371 | 0.229 | 0.410 | $0.403 \pm 0.117$ |
| Raincoat | **0.866** | 0.465 | 0.511 | 0.578 | 0.461 | $0.576 \pm 0.151$ |
| VRADA | 0.596 | 0.804 | 0.512 | 0.410 | 0.503 | $0.565 \pm 0.133$ |
| Target only | 0.952 | 0.947 | 0.951 | 0.937 | 0.948 | $0.947 \pm 0.005$ |

Table 13: Accuracy results on hhar dataset using Source Risk for the model selection and evaluated on Target Test

| | $e \rightarrow f$ | $g \rightarrow h$ | $c \rightarrow d$ | $i \rightarrow a$ | $a \rightarrow b$ | Avg |
|---|---|---|---|---|---|---|
| Source only | 0.708 | 0.506 | 0.573 | 0.412 | 0.504 | $0.541 \pm 0.098$ |
| CoDATS | 0.766 | 0.471 | **0.755** | 0.511 | 0.668 | $0.634 \pm 0.122$ |
| CoTMix | 0.875 | 0.591 | 0.744 | 0.580 | 0.687 | **0.695** $\pm$ **0.109** |
| InceptionCDAN | 0.687 | 0.391 | 0.619 | 0.456 | 0.446 | $0.52 \pm 0.113$ |
| InceptionDANN | **0.855** | 0.535 | 0.679 | 0.595 | 0.670 | $0.667 \pm 0.108$ |
| InceptionMix | **0.850** | 0.666 | 0.652 | **0.672** | 0.622 | **0.692** $\pm$ **0.081** |
| InceptionRain | **0.853** | 0.612 | 0.710 | 0.646 | 0.462 | $0.657 \pm 0.128$ |
| OTDA | 0.617 | 0.395 | 0.447 | 0.387 | 0.442 | $0.458 \pm 0.083$ |
| Raincoat | 0.793 | 0.448 | 0.482 | 0.481 | 0.475 | $0.536 \pm 0.129$ |
| VRADA | 0.617 | **0.770** | 0.564 | 0.538 | **0.712** | $0.64 \pm 0.088$ |
| Target only | 0.952 | 0.947 | 0.951 | 0.937 | 0.948 | $0.947 \pm 0.005$ |

Table 14: Accuracy results on hhar dataset using Target Risk for the model selection and evaluated on Target Test

| | $e \to f$ | $g \to h$ | $c \to d$ | $i \to a$ | $a \to b$ | Avg |
|---|---|---|---|---|---|---|
| Source only | 0.798 | 0.666 | 0.593 | 0.554 | 0.637 | $0.65 \pm 0.083$ |
| CoDATS | **0.916** | **0.943** | **0.810** | **0.777** | 0.700 | **0.829 $\pm$ 0.09** |
| CoTMix | 0.894 | 0.644 | 0.748 | 0.601 | **0.756** | $0.729 \pm 0.102$ |
| InceptionCDAN | 0.883 | 0.712 | 0.691 | 0.618 | 0.684 | $0.718 \pm 0.088$ |
| InceptionDANN | 0.857 | 0.627 | 0.732 | **0.776** | 0.686 | $0.736 \pm 0.078$ |
| InceptionMix | 0.877 | 0.917 | 0.565 | 0.686 | 0.694 | $0.748 \pm 0.131$ |
| InceptionRain | 0.886 | 0.908 | 0.720 | 0.665 | 0.603 | $0.756 \pm 0.121$ |
| OTDA | 0.577 | 0.448 | 0.442 | 0.387 | 0.422 | $0.455 \pm 0.065$ |
| Raincoat | **0.913** | 0.921 | 0.752 | 0.705 | 0.674 | $0.793 \pm 0.104$ |
| VRADA | 0.809 | 0.542 | 0.581 | 0.590 | 0.682 | $0.641 \pm 0.096$ |
| Target only | 0.952 | 0.947 | 0.951 | 0.937 | 0.948 | $0.947 \pm 0.005$ |

Table 15: Accuracy results on mfd dataset using IWCV for the model selection and evaluated on Target Test

| | $1 \to 2$ | $0 \to 1$ | $1 \to 3$ | $1 \to 0$ | $0 \to 3$ | Avg |
|---|---|---|---|---|---|---|
| Source only | 0.684 | 0.454 | 0.993 | 0.440 | 0.523 | $0.619 \pm 0.206$ |
| CoDATS | 0.754 | 0.569 | 0.951 | 0.173 | 0.454 | $0.58 \pm 0.264$ |
| CoTMix | 0.675 | 0.555 | 0.922 | 0.534 | 0.532 | $0.644 \pm 0.149$ |
| InceptionCDAN | **0.792** | 0.455 | 0.988 | **0.688** | **0.669** | **0.718 $\pm$ 0.174** |
| InceptionDANN | 0.734 | 0.454 | **0.992** | 0.366 | 0.472 | $0.604 \pm 0.23$ |
| InceptionMix | 0.507 | 0.573 | 0.950 | 0.445 | 0.580 | $0.611 \pm 0.176$ |
| InceptionRain | **0.799** | **0.717** | **0.993** | 0.450 | 0.493 | $0.69 \pm 0.2$ |
| OTDA | 0.478 | 0.474 | 0.528 | 0.476 | 0.476 | $0.486 \pm 0.021$ |
| Raincoat | 0.745 | 0.454 | **0.997** | 0.497 | 0.454 | $0.629 \pm 0.213$ |
| VRADA | 0.454 | 0.454 | 0.454 | 0.454 | 0.454 | $0.454 \pm 0.0$ |
| Target only | 1.000 | 1.000 | 1.000 | 0.993 | 1.000 | $0.999 \pm 0.003$ |

Table 16: Accuracy results on mfd dataset using Source Risk for the model selection and evaluated on Target Test

| | $1 \to 2$ | $0 \to 1$ | $1 \to 3$ | $1 \to 0$ | $0 \to 3$ | Avg |
|---|---|---|---|---|---|---|
| Source only | 0.777 | 0.491 | 0.958 | 0.371 | 0.518 | $0.623 \pm 0.213$ |
| CoDATS | 0.735 | 0.455 | 0.947 | 0.315 | 0.454 | $0.581 \pm 0.228$ |
| CoTMix | 0.605 | 0.565 | 0.954 | 0.487 | 0.570 | $0.636 \pm 0.164$ |
| InceptionCDAN | 0.789 | 0.468 | 0.946 | **0.576** | **0.590** | $0.674 \pm 0.171$ |
| InceptionDANN | 0.821 | 0.454 | 0.994 | 0.471 | 0.454 | $0.639 \pm 0.226$ |
| InceptionMix | 0.656 | 0.555 | 0.991 | 0.555 | 0.505 | $0.652 \pm 0.176$ |
| InceptionRain | **0.902** | **0.700** | **1.000** | 0.350 | 0.635 | **0.717 $\pm$ 0.226** |
| OTDA | 0.492 | 0.481 | 0.534 | 0.430 | 0.485 | $0.484 \pm 0.033$ |
| Raincoat | 0.716 | 0.454 | 0.977 | 0.302 | 0.444 | $0.579 \pm 0.24$ |
| VRADA | 0.456 | 0.447 | 0.461 | 0.455 | 0.455 | $0.455 \pm 0.004$ |
| Target only | 1.000 | 1.000 | 1.000 | 0.993 | 1.000 | $0.999 \pm 0.003$ |

Table 17: Accuracy results on mfd dataset using Target Risk for the model selection and evaluated on Target Test

| | $1 \to 2$ | $0 \to 1$ | $1 \to 3$ | $1 \to 0$ | $0 \to 3$ | Avg |
|---|---|---|---|---|---|---|
| Source only | 0.795 | 0.582 | 0.999 | 0.688 | 0.495 | $0.712 \pm 0.175$ |
| CoDATS | 0.719 | 0.584 | 0.989 | 0.634 | 0.686 | $0.722 \pm 0.141$ |
| CoTMix | 0.695 | 0.612 | 0.922 | 0.670 | 0.586 | $0.697 \pm 0.119$ |
| InceptionCDAN | 0.896 | 0.670 | **0.997** | **0.880** | 0.544 | $0.797 \pm 0.165$ |
| InceptionDANN | 0.836 | **0.956** | **0.997** | 0.455 | **0.808** | $\mathbf{0.81 \pm 0.191}$ |
| InceptionMix | 0.738 | 0.779 | **0.997** | 0.645 | 0.735 | $0.779 \pm 0.118$ |
| InceptionRain | 0.865 | 0.729 | **0.999** | **0.889** | 0.744 | $0.845 \pm 0.1$ |
| OTDA | 0.469 | 0.472 | 0.475 | 0.528 | 0.448 | $0.478 \pm 0.027$ |
| Raincoat | **0.908** | 0.476 | 0.974 | 0.657 | 0.580 | $0.719 \pm 0.191$ |
| VRADA | 0.454 | 0.447 | 0.471 | 0.440 | 0.455 | $0.453 \pm 0.01$ |
| Target only | 1.000 | 1.000 | 1.000 | 0.993 | 1.000 | $0.999 \pm 0.003$ |

Table 18: Accuracy results on miniTimeMatch dataset using IWCV for the model selection and evaluated on Target Test

| | $Fr_2 \to Dk_1$ | $Fr_1 \to At_1$ | $Dk_1 \to Fr_1$ | $At_1 \to Fr_2$ | $Fr_1 \to Fr_2$ | Avg |
|---|---|---|---|---|---|---|
| Source only | 0.292 | 0.845 | 0.711 | 0.668 | 0.861 | $0.675 \pm 0.206$ |
| CoDATS | 0.435 | 0.658 | 0.720 | 0.780 | 0.833 | $0.685 \pm 0.138$ |
| CoTMix | 0.266 | 0.266 | 0.470 | 0.501 | 0.333 | $0.367 \pm 0.1$ |
| InceptionCDAN | 0.793 | 0.888 | 0.693 | **0.918** | 0.841 | $0.827 \pm 0.079$ |
| InceptionDANN | **0.828** | 0.929 | 0.741 | 0.888 | 0.880 | $0.853 \pm 0.065$ |
| InceptionMix | 0.140 | 0.140 | 0.663 | 0.140 | 0.140 | $0.245 \pm 0.209$ |
| InceptionRain | 0.699 | **0.950** | **0.847** | 0.914 | **0.915** | $\mathbf{0.865 \pm 0.089}$ |
| OTDA | 0.448 | 0.439 | 0.439 | 0.439 | 0.439 | $0.441 \pm 0.004$ |
| Raincoat | 0.445 | 0.875 | 0.676 | 0.840 | 0.908 | $0.749 \pm 0.172$ |
| VRADA | 0.469 | 0.435 | 0.518 | 0.735 | 0.807 | $0.593 \pm 0.15$ |
| Target only | 0.896 | 0.672 | 0.838 | 0.976 | 0.976 | $0.872 \pm 0.113$ |

Table 19: Accuracy results on miniTimeMatch dataset using Source Risk for the model selection and evaluated on Target Test

| | $Fr_2 \to Dk_1$ | $Fr_1 \to At_1$ | $Dk_1 \to Fr_1$ | $At_1 \to Fr_2$ | $Fr_1 \to Fr_2$ | Avg |
|---|---|---|---|---|---|---|
| Source only | 0.588 | 0.671 | 0.749 | 0.856 | 0.866 | $0.746 \pm 0.107$ |
| CoDATS | 0.721 | 0.654 | 0.578 | 0.769 | 0.822 | $0.709 \pm 0.086$ |
| CoTMix | 0.378 | 0.241 | 0.501 | 0.638 | 0.426 | $0.437 \pm 0.132$ |
| InceptionCDAN | **0.838** | 0.943 | 0.713 | **0.931** | **0.948** | $\mathbf{0.875 \pm 0.09}$ |
| InceptionDANN | 0.807 | 0.936 | 0.796 | 0.913 | 0.883 | $0.867 \pm 0.056$ |
| InceptionMix | 0.140 | 0.140 | 0.140 | 0.140 | 0.140 | $0.14 \pm 0.0$ |
| InceptionRain | 0.783 | **0.953** | **0.805** | 0.733 | 0.929 | $0.841 \pm 0.086$ |
| OTDA | 0.439 | 0.439 | 0.439 | 0.268 | 0.439 | $0.405 \pm 0.068$ |
| Raincoat | 0.763 | 0.947 | 0.648 | 0.870 | 0.892 | $0.824 \pm 0.106$ |
| VRADA | 0.459 | 0.545 | 0.566 | 0.731 | 0.819 | $0.624 \pm 0.131$ |
| Target only | 0.896 | 0.672 | 0.838 | 0.976 | 0.976 | $0.872 \pm 0.113$ |

Table 20: Accuracy results on miniTimeMatch dataset using Target Risk for the model selection and evaluated on Target Test

| | $Fr_2 \to Dk_1$ | $Fr_1 \to At_1$ | $Dk_1 \to Fr_1$ | $At_1 \to Fr_2$ | $Fr_1 \to Fr_2$ | Avg |
|---|---|---|---|---|---|---|
| Source only | 0.604 | 0.900 | 0.777 | 0.829 | 0.860 | $0.794 \pm 0.103$ |
| CoDATS | 0.729 | 0.832 | 0.668 | 0.897 | 0.906 | $0.806 \pm 0.094$ |
| CoTMix | 0.364 | 0.215 | 0.529 | 0.497 | 0.468 | $0.415 \pm 0.114$ |
| InceptionCDAN | 0.846 | 0.949 | **0.830** | 0.928 | 0.948 | $0.9 \pm 0.052$ |
| InceptionDANN | **0.855** | 0.956 | 0.779 | **0.931** | 0.949 | $\mathbf{0.894 \pm 0.068}$ |
| InceptionMix | 0.140 | 0.140 | 0.613 | 0.140 | 0.140 | $0.235 \pm 0.189$ |
| InceptionRain | **0.853** | **0.957** | 0.679 | **0.933** | **0.952** | $0.875 \pm 0.105$ |
| OTDA | 0.439 | 0.466 | 0.365 | 0.602 | 0.645 | $0.503 \pm 0.104$ |
| Raincoat | 0.787 | 0.943 | 0.718 | 0.893 | 0.912 | $0.851 \pm 0.085$ |
| VRADA | 0.504 | 0.540 | 0.468 | 0.700 | 0.790 | $0.6 \pm 0.124$ |
| Target only | 0.896 | 0.672 | 0.838 | 0.976 | 0.976 | $0.872 \pm 0.113$ |

Table 21: Accuracy results on OnHWeq dataset using IWCV for the model selection and evaluated on Target Test

| | $L \to R$ | $R \to L$ | Avg |
|---|---|---|---|
| Source only | 0.389 | 0.486 | $0.438 \pm 0.048$ |
| CoDATS | **0.633** | **0.583** | $\mathbf{0.608 \pm 0.025}$ |
| CoTMix | 0.357 | 0.334 | $0.346 \pm 0.011$ |
| InceptionCDAN | 0.353 | 0.494 | $0.424 \pm 0.07$ |
| InceptionDANN | 0.526 | 0.495 | $0.51 \pm 0.016$ |
| InceptionMix | 0.522 | 0.435 | $0.479 \pm 0.044$ |
| InceptionRain | 0.512 | 0.503 | $0.508 \pm 0.005$ |
| OTDA | 0.419 | 0.431 | $0.425 \pm 0.006$ |
| Raincoat | 0.499 | 0.514 | $0.506 \pm 0.008$ |
| VRADA | 0.348 | 0.454 | $0.401 \pm 0.053$ |
| Target only | 0.933 | 0.932 | $0.933 \pm 0.001$ |

Table 22: Accuracy results on OnHWeq dataset using Source Risk for the model selection and evaluated on Target Test

| | $L \to R$ | $R \to L$ | Avg |
|---|---|---|---|
| Source only | 0.423 | 0.510 | $0.466 \pm 0.044$ |
| CoDATS | 0.552 | 0.476 | $0.514 \pm 0.038$ |
| CoTMix | 0.393 | 0.323 | $0.358 \pm 0.035$ |
| InceptionCDAN | 0.322 | 0.492 | $0.407 \pm 0.085$ |
| InceptionDANN | 0.415 | 0.506 | $0.46 \pm 0.046$ |
| InceptionMix | 0.547 | 0.459 | $0.503 \pm 0.044$ |
| InceptionRain | 0.540 | 0.487 | $0.514 \pm 0.027$ |
| OTDA | 0.431 | 0.431 | $0.431 \pm 0.0$ |
| Raincoat | **0.563** | **0.525** | $\mathbf{0.544 \pm 0.019}$ |
| VRADA | 0.375 | 0.454 | $0.414 \pm 0.04$ |
| Target only | 0.933 | 0.932 | $0.933 \pm 0.001$ |

Table 23: Accuracy results on OnHWeq dataset using Target Risk for the model selection and evaluated on Target Test

|  | $L \to R$ | $R \to L$ | Avg |
|---|---|---|---|
| Source only | 0.437 | 0.513 | $0.475 \pm 0.038$ |
| CoDATS | 0.667 | 0.584 | $0.626 \pm 0.042$ |
| CoTMix | 0.441 | 0.334 | $0.388 \pm 0.053$ |
| InceptionCDAN | 0.660 | 0.492 | $0.576 \pm 0.084$ |
| InceptionDANN | **0.676** | 0.526 | $0.601 \pm 0.075$ |
| InceptionMix | 0.540 | 0.458 | $0.499 \pm 0.041$ |
| InceptionRain | **0.677** | **0.613** | $\mathbf{0.645 \pm 0.032}$ |
| OTDA | 0.457 | 0.431 | $0.444 \pm 0.013$ |
| Raincoat | 0.580 | 0.546 | $0.563 \pm 0.017$ |
| VRADA | 0.362 | 0.438 | $0.4 \pm 0.038$ |
| Target only | 0.933 | 0.932 | $0.933 \pm 0.001$ |

Table 24: Accuracy results on ptbXLecg dataset using IWCV for the model selection and evaluated on Target Test

|  | $0 \to 1$ | $0 \to 3$ | $2 \to 1$ | $1 \to 3$ | $0 \to 2$ | Avg |
|---|---|---|---|---|---|---|
| Source only | 0.690 | 0.740 | 0.654 | 0.623 | 0.645 | $0.67 \pm 0.041$ |
| CoDATS | 0.572 | 0.623 | 0.585 | 0.597 | 0.569 | $0.589 \pm 0.02$ |
| CoTMix | 0.507 | 0.649 | 0.607 | 0.597 | 0.640 | $0.6 \pm 0.05$ |
| InceptionCDAN | **0.686** | 0.636 | 0.672 | 0.610 | **0.695** | $\mathbf{0.66 \pm 0.032}$ |
| InceptionDANN | 0.644 | 0.571 | 0.609 | 0.636 | 0.614 | $0.615 \pm 0.026$ |
| InceptionMix | 0.594 | **0.675** | **0.688** | 0.636 | 0.613 | $0.641 \pm 0.036$ |
| InceptionRain | 0.644 | 0.442 | 0.663 | **0.649** | 0.654 | $0.61 \pm 0.084$ |
| OTDA | 0.585 | 0.416 | 0.585 | 0.416 | 0.513 | $0.503 \pm 0.076$ |
| Raincoat | 0.584 | 0.519 | 0.505 | 0.571 | 0.550 | $0.546 \pm 0.03$ |
| VRADA | 0.605 | 0.571 | **0.681** | 0.558 | 0.594 | $0.602 \pm 0.043$ |
| Target only | 0.699 | 0.636 | 0.704 | 0.571 | 0.675 | $0.657 \pm 0.049$ |

Table 25: Accuracy results on ptbXLecg dataset using Source Risk for the model selection and evaluated on Target Test

|  | $0 \to 1$ | $0 \to 3$ | $2 \to 1$ | $1 \to 3$ | $0 \to 2$ | Avg |
|---|---|---|---|---|---|---|
| Source only | 0.668 | 0.649 | 0.645 | 0.675 | 0.639 | $0.655 \pm 0.014$ |
| CoDATS | 0.681 | 0.688 | 0.583 | 0.506 | 0.618 | $0.615 \pm 0.067$ |
| CoTMix | 0.454 | 0.649 | 0.581 | 0.623 | 0.645 | $0.59 \pm 0.072$ |
| InceptionCDAN | **0.709** | 0.623 | 0.660 | 0.584 | **0.674** | $\mathbf{0.65 \pm 0.043}$ |
| InceptionDANN | 0.607 | **0.727** | **0.700** | 0.597 | 0.664 | $0.659 \pm 0.051$ |
| InceptionMix | 0.611 | 0.675 | 0.691 | 0.584 | 0.665 | $0.645 \pm 0.041$ |
| InceptionRain | 0.686 | 0.610 | 0.683 | **0.623** | **0.670** | $\mathbf{0.654 \pm 0.032}$ |
| OTDA | 0.585 | 0.416 | 0.585 | 0.416 | 0.513 | $0.503 \pm 0.076$ |
| Raincoat | 0.659 | 0.597 | 0.517 | 0.571 | 0.607 | $0.59 \pm 0.046$ |
| VRADA | 0.587 | 0.571 | 0.677 | 0.545 | 0.582 | $0.592 \pm 0.045$ |
| Target only | 0.699 | 0.636 | 0.704 | 0.571 | 0.675 | $0.657 \pm 0.049$ |

Table 26: Accuracy results on ptbXLecg dataset using Target Risk for the model selection and evaluated on Target Test

| | $0 \to 1$ | $0 \to 3$ | $2 \to 1$ | $1 \to 3$ | $0 \to 2$ | Avg |
|---|---|---|---|---|---|---|
| Source only | 0.696 | 0.636 | 0.653 | 0.675 | 0.673 | $0.667 \pm 0.02$ |
| CoDATS | 0.660 | 0.636 | 0.654 | 0.610 | 0.632 | $0.638 \pm 0.018$ |
| CoTMix | 0.585 | 0.571 | 0.638 | 0.623 | 0.641 | $0.612 \pm 0.028$ |
| InceptionCDAN | 0.698 | 0.688 | 0.672 | **0.688** | **0.685** | **$0.686 \pm 0.008$** |
| InceptionDANN | **0.700** | **0.701** | **0.693** | 0.545 | 0.644 | $0.657 \pm 0.06$ |
| InceptionMix | 0.695 | 0.688 | 0.665 | 0.623 | 0.658 | $0.666 \pm 0.025$ |
| InceptionRain | 0.660 | 0.597 | 0.685 | 0.610 | 0.630 | $0.636 \pm 0.032$ |
| OTDA | 0.581 | 0.364 | 0.586 | 0.364 | 0.514 | $0.482 \pm 0.099$ |
| Raincoat | 0.657 | 0.494 | 0.605 | 0.584 | 0.590 | $0.586 \pm 0.053$ |
| VRADA | 0.601 | 0.558 | 0.672 | 0.545 | 0.560 | $0.587 \pm 0.046$ |
| Target only | 0.699 | 0.636 | 0.704 | 0.571 | 0.675 | $0.657 \pm 0.049$ |

Table 27: Accuracy results on sleepStage dataset using IWCV for the model selection and evaluated on Target Test

| | $12 \to 5$ | $0 \to 11$ | $16 \to 1$ | $7 \to 18$ | $9 \to 14$ | Avg |
|---|---|---|---|---|---|---|
| Source only | 0.300 | 0.466 | 0.571 | 0.530 | 0.680 | $0.509 \pm 0.126$ |
| CoDATS | 0.788 | 0.563 | 0.635 | 0.519 | **0.811** | $0.663 \pm 0.118$ |
| CoTMix | 0.533 | 0.489 | 0.555 | 0.754 | 0.772 | $0.621 \pm 0.118$ |
| InceptionCDAN | **0.861** | 0.570 | 0.553 | **0.802** | **0.814** | **$0.72 \pm 0.131$** |
| InceptionDANN | 0.615 | 0.592 | 0.624 | 0.746 | 0.784 | $0.672 \pm 0.077$ |
| InceptionMix | 0.665 | 0.539 | 0.703 | 0.784 | 0.734 | $0.685 \pm 0.083$ |
| InceptionRain | 0.717 | **0.640** | **0.729** | 0.716 | 0.780 | $0.716 \pm 0.045$ |
| OTDA | 0.241 | 0.518 | 0.534 | 0.360 | 0.404 | $0.411 \pm 0.108$ |
| Raincoat | 0.811 | 0.566 | 0.658 | 0.777 | 0.726 | $0.708 \pm 0.088$ |
| VRADA | 0.457 | 0.455 | 0.571 | 0.479 | 0.552 | $0.503 \pm 0.049$ |
| Target only | 0.868 | 0.881 | 0.859 | 0.848 | 0.944 | $0.88 \pm 0.034$ |

Table 28: Accuracy results on sleepStage dataset using Source Risk for the model selection and evaluated on Target Test

| | $12 \to 5$ | $0 \to 11$ | $16 \to 1$ | $7 \to 18$ | $9 \to 14$ | Avg |
|---|---|---|---|---|---|---|
| Source only | 0.736 | 0.495 | 0.569 | 0.583 | 0.683 | $0.613 \pm 0.086$ |
| CoDATS | 0.726 | 0.601 | **0.793** | 0.636 | 0.833 | $0.718 \pm 0.089$ |
| CoTMix | 0.736 | 0.574 | 0.729 | 0.693 | 0.664 | $0.679 \pm 0.059$ |
| InceptionCDAN | 0.767 | 0.653 | 0.480 | **0.807** | **0.843** | $0.71 \pm 0.132$ |
| InceptionDANN | 0.780 | 0.661 | 0.480 | 0.754 | 0.779 | $0.691 \pm 0.114$ |
| InceptionMix | **0.806** | 0.401 | 0.409 | 0.769 | 0.719 | $0.621 \pm 0.178$ |
| InceptionRain | 0.712 | 0.570 | 0.764 | 0.777 | **0.847** | **$0.734 \pm 0.093$** |
| OTDA | 0.434 | 0.518 | 0.534 | 0.314 | 0.399 | $0.44 \pm 0.081$ |
| Raincoat | 0.726 | **0.632** | 0.639 | 0.772 | 0.656 | $0.685 \pm 0.055$ |
| VRADA | 0.552 | 0.430 | 0.582 | 0.479 | 0.455 | $0.5 \pm 0.058$ |
| Target only | 0.868 | 0.881 | 0.859 | 0.848 | 0.944 | $0.88 \pm 0.034$ |

Table 29: Accuracy results on sleepStage dataset using Target Risk for the model selection and evaluated on Target Test

|  | $12 \rightarrow 5$ | $0 \rightarrow 11$ | $16 \rightarrow 1$ | $7 \rightarrow 18$ | $9 \rightarrow 14$ | Avg |
|---|---|---|---|---|---|---|
| Source only | 0.795 | 0.526 | 0.697 | 0.760 | 0.836 | $0.723 \pm 0.108$ |
| CoDATS | **0.833** | 0.694 | 0.777 | 0.769 | 0.826 | **0.78 ± 0.05** |
| CoTMix | 0.799 | 0.649 | 0.799 | 0.746 | 0.744 | $0.747 \pm 0.055$ |
| InceptionCDAN | **0.832** | **0.825** | **0.806** | 0.786 | 0.678 | **0.785 ± 0.056** |
| InceptionDANN | 0.778 | 0.726 | 0.742 | **0.799** | **0.847** | $0.778 \pm 0.043$ |
| InceptionMix | 0.743 | 0.559 | 0.735 | 0.746 | 0.767 | $0.71 \pm 0.076$ |
| InceptionRain | **0.835** | 0.759 | 0.796 | 0.746 | 0.753 | $0.778 \pm 0.033$ |
| OTDA | 0.434 | 0.518 | 0.534 | 0.339 | 0.390 | $0.443 \pm 0.074$ |
| Raincoat | 0.802 | 0.682 | 0.769 | 0.774 | 0.792 | $0.764 \pm 0.043$ |
| VRADA | 0.561 | 0.464 | 0.584 | 0.564 | 0.525 | $0.54 \pm 0.042$ |
| Target only | 0.868 | 0.881 | 0.859 | 0.848 | 0.944 | $0.88 \pm 0.034$ |

Table 30: Accuracy results on sportsActivities dataset using IWCV for the model selection and evaluated on Target Test

|  | $p7 \rightarrow p3$ | $p4 \rightarrow p2$ | $p1 \rightarrow p8$ | $p5 \rightarrow p4$ | $p5 \rightarrow p6$ | Avg |
|---|---|---|---|---|---|---|
| Source only | 0.596 | 0.610 | 0.496 | 0.662 | 0.364 | $0.546 \pm 0.106$ |
| CoDATS | **0.820** | 0.632 | 0.535 | 0.794 | 0.399 | $0.636 \pm 0.158$ |
| CoTMix | 0.219 | 0.206 | 0.285 | 0.355 | 0.246 | $0.262 \pm 0.054$ |
| InceptionCDAN | 0.588 | 0.316 | 0.404 | 0.675 | **0.697** | $0.536 \pm 0.151$ |
| InceptionDANN | 0.706 | 0.500 | 0.469 | 0.711 | 0.548 | $0.587 \pm 0.103$ |
| InceptionMix | 0.053 | 0.539 | 0.482 | 0.053 | 0.053 | $0.236 \pm 0.225$ |
| InceptionRain | 0.434 | **0.846** | 0.478 | 0.711 | 0.412 | $0.576 \pm 0.172$ |
| OTDA | 0.307 | 0.526 | 0.531 | 0.469 | 0.535 | $0.474 \pm 0.087$ |
| Raincoat | 0.596 | 0.706 | **0.702** | **0.820** | 0.566 | **0.678 ± 0.09** |
| VRADA | 0.346 | 0.368 | 0.232 | 0.482 | 0.360 | $0.358 \pm 0.079$ |
| Target only | 0.991 | 1.000 | 0.978 | 0.991 | 0.987 | $0.989 \pm 0.007$ |

Table 31: Accuracy results on sportsActivities dataset using Source Risk for the model selection and evaluated on Target Test

|  | $p7 \rightarrow p3$ | $p4 \rightarrow p2$ | $p1 \rightarrow p8$ | $p5 \rightarrow p4$ | $p5 \rightarrow p6$ | Avg |
|---|---|---|---|---|---|---|
| Source only | 0.548 | 0.557 | 0.504 | 0.592 | 0.439 | $0.528 \pm 0.053$ |
| CoDATS | **0.768** | 0.759 | 0.482 | 0.539 | **0.610** | **0.632 ± 0.115** |
| CoTMix | 0.254 | 0.360 | 0.289 | 0.421 | 0.316 | $0.328 \pm 0.058$ |
| InceptionCDAN | 0.737 | 0.478 | 0.333 | 0.518 | 0.360 | $0.485 \pm 0.144$ |
| InceptionDANN | 0.544 | **0.803** | 0.443 | **0.728** | 0.500 | $0.604 \pm 0.138$ |
| InceptionMix | 0.053 | 0.772 | 0.509 | 0.053 | 0.053 | $0.288 \pm 0.3$ |
| InceptionRain | 0.627 | 0.636 | 0.526 | 0.662 | 0.478 | $0.586 \pm 0.071$ |
| OTDA | 0.298 | 0.526 | 0.557 | 0.456 | 0.553 | $0.478 \pm 0.097$ |
| Raincoat | 0.408 | 0.588 | **0.575** | 0.719 | 0.259 | $0.51 \pm 0.16$ |
| VRADA | 0.298 | 0.368 | 0.232 | 0.206 | 0.285 | $0.278 \pm 0.056$ |
| Target only | 0.991 | 1.000 | 0.978 | 0.991 | 0.987 | $0.989 \pm 0.007$ |

Table 32: Accuracy results on sportsActivities dataset using Target Risk for the model selection and evaluated on Target Test

| | $p7 \rightarrow p3$ | $p4 \rightarrow p2$ | $p1 \rightarrow p8$ | $p5 \rightarrow p4$ | $p5 \rightarrow p6$ | Avg |
|---|---|---|---|---|---|---|
| Source only | 0.596 | 0.675 | 0.526 | 0.838 | 0.623 | $0.652 \pm 0.105$ |
| CoDATS | **0.921** | 0.851 | 0.724 | **0.886** | 0.789 | $0.834 \pm 0.07$ |
| CoTMix | 0.250 | 0.307 | 0.325 | 0.443 | 0.285 | $0.322 \pm 0.065$ |
| InceptionCDAN | 0.741 | 0.746 | 0.592 | 0.825 | 0.557 | $0.692 \pm 0.101$ |
| InceptionDANN | 0.886 | 0.754 | 0.684 | 0.789 | 0.697 | $0.762 \pm 0.073$ |
| InceptionMix | 0.053 | 0.776 | 0.434 | 0.053 | 0.053 | $0.274 \pm 0.291$ |
| InceptionRain | 0.711 | 0.636 | 0.632 | 0.728 | 0.610 | $0.663 \pm 0.047$ |
| OTDA | 0.289 | 0.526 | 0.531 | 0.461 | 0.535 | $0.468 \pm 0.094$ |
| Raincoat | 0.768 | **0.860** | **0.895** | 0.864 | **0.912** | **$0.86 \pm 0.05$** |
| VRADA | 0.478 | 0.325 | 0.250 | 0.776 | 0.268 | $0.419 \pm 0.196$ |
| Target only | 0.991 | 1.000 | 0.978 | 0.991 | 0.987 | $0.989 \pm 0.007$ |

Table 33: Accuracy results on ultrasoundMuscleContraction dataset using IWCV for the model selection and evaluated on Target Test

| | $sb1 \rightarrow sb8$ | $sb8 \rightarrow sb6$ | $sb2 \rightarrow sb7$ | $sb5 \rightarrow sb4$ | $sb3 \rightarrow sb5$ | Avg |
|---|---|---|---|---|---|---|
| Source only | 0.365 | 0.396 | 0.539 | 0.460 | 0.651 | $0.482 \pm 0.103$ |
| CoDATS | **0.581** | 0.355 | 0.461 | 0.508 | 0.359 | $0.453 \pm 0.087$ |
| CoTMix | 0.533 | 0.486 | 0.500 | 0.422 | **0.718** | $0.532 \pm 0.1$ |
| InceptionCDAN | 0.330 | **0.645** | 0.539 | 0.493 | 0.359 | $0.473 \pm 0.116$ |
| InceptionDANN | 0.419 | 0.355 | 0.461 | 0.481 | 0.641 | $0.471 \pm 0.095$ |
| InceptionMix | 0.513 | 0.271 | **0.645** | 0.456 | 0.465 | $0.47 \pm 0.12$ |
| InceptionRain | 0.350 | **0.645** | 0.539 | **0.513** | 0.623 | $0.534 \pm 0.104$ |
| OTDA | **0.581** | **0.645** | 0.539 | 0.460 | 0.641 | $0.573 \pm 0.069$ |
| Raincoat | 0.532 | **0.645** | 0.470 | 0.472 | 0.515 | $0.527 \pm 0.064$ |
| VRADA | **0.581** | **0.645** | 0.539 | 0.508 | 0.641 | **$0.583 \pm 0.054$** |
| Target only | 0.980 | 0.993 | 0.939 | 0.980 | 0.982 | $0.975 \pm 0.019$ |

Table 34: Accuracy results on ultrasoundMuscleContraction dataset using Source Risk for the model selection and evaluated on Target Test

| | $sb1 \rightarrow sb8$ | $sb8 \rightarrow sb6$ | $sb2 \rightarrow sb7$ | $sb5 \rightarrow sb4$ | $sb3 \rightarrow sb5$ | Avg |
|---|---|---|---|---|---|---|
| Source only | 0.413 | 0.274 | 0.451 | 0.488 | 0.359 | $0.397 \pm 0.075$ |
| CoDATS | 0.584 | 0.352 | 0.444 | 0.554 | 0.406 | $0.468 \pm 0.088$ |
| CoTMix | **0.595** | 0.488 | 0.512 | **0.561** | 0.502 | $0.532 \pm 0.04$ |
| InceptionCDAN | 0.419 | 0.347 | 0.465 | 0.445 | 0.389 | $0.413 \pm 0.042$ |
| InceptionDANN | 0.414 | 0.368 | 0.566 | 0.442 | 0.506 | $0.459 \pm 0.07$ |
| InceptionMix | 0.331 | 0.420 | **0.818** | 0.423 | 0.516 | $0.502 \pm 0.169$ |
| InceptionRain | 0.556 | 0.292 | 0.601 | 0.510 | 0.417 | $0.475 \pm 0.11$ |
| OTDA | 0.581 | **0.645** | 0.484 | 0.508 | 0.438 | $0.531 \pm 0.073$ |
| Raincoat | 0.403 | 0.299 | 0.472 | 0.476 | 0.359 | $0.402 \pm 0.068$ |
| VRADA | 0.581 | **0.645** | 0.539 | 0.508 | **0.641** | **$0.583 \pm 0.054$** |
| Target only | 0.980 | 0.993 | 0.939 | 0.980 | 0.982 | $0.975 \pm 0.019$ |

Table 35: Accuracy results on ultrasoundMuscleContraction dataset using Target Risk for the model selection and evaluated on Target Test

|  | $sb1 \rightarrow sb8$ | $sb8 \rightarrow sb6$ | $sb2 \rightarrow sb7$ | $sb5 \rightarrow sb4$ | $sb3 \rightarrow sb5$ | Avg |
|---|---|---|---|---|---|---|
| Source only | 0.419 | 0.361 | 0.827 | 0.540 | 0.593 | $0.548 \pm 0.162$ |
| CoDATS | 0.647 | 0.645 | 0.783 | 0.572 | 0.644 | $0.658 \pm 0.069$ |
| CoTMix | **0.722** | **0.680** | **0.812** | 0.529 | **0.793** | **0.707 ± 0.101** |
| InceptionCDAN | 0.581 | 0.649 | 0.790 | 0.598 | 0.359 | $0.595 \pm 0.139$ |
| InceptionDANN | 0.581 | 0.645 | 0.843 | 0.581 | 0.641 | $0.658 \pm 0.096$ |
| InceptionMix | 0.517 | 0.352 | 0.753 | 0.501 | 0.478 | $0.52 \pm 0.13$ |
| InceptionRain | 0.586 | 0.645 | 0.624 | 0.557 | 0.619 | $0.606 \pm 0.031$ |
| OTDA | 0.581 | 0.645 | 0.578 | **0.604** | 0.641 | $0.61 \pm 0.029$ |
| Raincoat | 0.664 | 0.648 | 0.649 | 0.593 | 0.670 | $0.645 \pm 0.027$ |
| VRADA | 0.581 | 0.645 | 0.539 | 0.508 | 0.641 | $0.583 \pm 0.054$ |
| Target only | 0.980 | 0.993 | 0.939 | 0.980 | 0.982 | $0.975 \pm 0.019$ |

Table 36: Accuracy results on wisdm dataset using IWCV for the model selection and evaluated on Target Test

|  | $3 \rightarrow 5$ | $27 \rightarrow 3$ | $5 \rightarrow 1$ | $7 \rightarrow 30$ | $21 \rightarrow 31$ | Avg |
|---|---|---|---|---|---|---|
| Source only | 0.804 | 0.621 | 0.042 | 0.171 | 0.451 | $0.418 \pm 0.281$ |
| CoDATS | 0.565 | 0.621 | 0.417 | 0.537 | 0.577 | $0.543 \pm 0.069$ |
| CoTMix | 0.609 | 0.448 | 0.792 | 0.463 | **0.718** | $0.606 \pm 0.136$ |
| InceptionCDAN | 0.630 | 0.707 | 0.417 | 0.512 | 0.662 | $0.586 \pm 0.106$ |
| InceptionDANN | 0.413 | 0.414 | 0.104 | **0.707** | 0.507 | $0.429 \pm 0.195$ |
| InceptionMix | 0.326 | 0.517 | 0.375 | 0.659 | 0.465 | $0.468 \pm 0.116$ |
| InceptionRain | 0.630 | **0.810** | **0.854** | 0.634 | 0.648 | **0.715 ± 0.097** |
| OTDA | 0.435 | 0.310 | 0.417 | 0.366 | 0.352 | $0.376 \pm 0.045$ |
| Raincoat | **0.739** | 0.672 | 0.729 | 0.244 | 0.549 | $0.587 \pm 0.184$ |
| VRADA | 0.239 | 0.293 | 0.667 | 0.488 | 0.380 | $0.413 \pm 0.152$ |
| Target only | 0.630 | 0.707 | 0.833 | 0.512 | 0.563 | $0.649 \pm 0.113$ |

Table 37: Accuracy results on wisdm dataset using Source Risk for the model selection and evaluated on Target Test

|  | $3 \rightarrow 5$ | $27 \rightarrow 3$ | $5 \rightarrow 1$ | $7 \rightarrow 30$ | $21 \rightarrow 31$ | Avg |
|---|---|---|---|---|---|---|
| Source only | 0.413 | 0.362 | 0.417 | 0.244 | 0.338 | $0.355 \pm 0.063$ |
| CoDATS | 0.696 | **0.793** | 0.417 | **0.439** | 0.423 | $0.554 \pm 0.159$ |
| CoTMix | 0.304 | 0.431 | 0.479 | 0.098 | 0.239 | $0.31 \pm 0.137$ |
| InceptionCDAN | 0.630 | 0.603 | 0.417 | 0.244 | 0.380 | $0.455 \pm 0.144$ |
| InceptionDANN | 0.609 | 0.328 | 0.583 | 0.098 | 0.507 | $0.425 \pm 0.191$ |
| InceptionMix | **0.761** | 0.259 | 0.021 | 0.049 | **0.718** | $0.362 \pm 0.32$ |
| InceptionRain | 0.630 | 0.776 | 0.417 | 0.000 | 0.394 | $0.443 \pm 0.263$ |
| OTDA | 0.348 | 0.293 | 0.417 | **0.439** | 0.380 | $0.375 \pm 0.052$ |
| Raincoat | 0.652 | **0.793** | **0.792** | 0.293 | **0.718** | **0.65 ± 0.186** |
| VRADA | 0.652 | 0.328 | 0.417 | 0.146 | 0.437 | $0.396 \pm 0.164$ |
| Target only | 0.630 | 0.707 | 0.833 | 0.512 | 0.563 | $0.649 \pm 0.113$ |

Table 38: Accuracy results on wisdm dataset using Target Risk for the model selection and evaluated on Target Test

|  | $3 \to 5$ | $27 \to 3$ | $5 \to 1$ | $7 \to 30$ | $21 \to 31$ | Avg |
|---|---|---|---|---|---|---|
| Source only | 0.413 | 0.741 | 0.417 | 0.561 | 0.775 | $0.581 \pm 0.154$ |
| CoDATS | 0.413 | 0.690 | 0.771 | 0.488 | 0.761 | $0.625 \pm 0.147$ |
| CoTMix | 0.630 | 0.672 | **0.833** | 0.610 | 0.775 | $0.704 \pm 0.086$ |
| InceptionCDAN | 0.413 | **0.931** | 0.792 | 0.488 | 0.859 | $0.697 \pm 0.207$ |
| InceptionDANN | 0.413 | 0.362 | 0.792 | **0.732** | 0.380 | $0.536 \pm 0.186$ |
| InceptionMix | **0.804** | 0.310 | 0.375 | 0.659 | 0.817 | $0.593 \pm 0.213$ |
| InceptionRain | 0.630 | 0.759 | 0.792 | 0.512 | **0.972** | $\mathbf{0.733 \pm 0.155}$ |
| OTDA | 0.413 | 0.362 | 0.583 | 0.488 | 0.380 | $0.445 \pm 0.081$ |
| Raincoat | 0.652 | 0.724 | 0.792 | 0.537 | 0.859 | $0.713 \pm 0.112$ |
| VRADA | 0.413 | 0.362 | 0.417 | 0.488 | 0.380 | $0.412 \pm 0.043$ |
| Target only | 0.630 | 0.707 | 0.833 | 0.512 | 0.563 | $0.649 \pm 0.113$ |

### A.1.2 AVERAGE PERFORMANCES: DATASETS AND CLASSIFIERS

In this section, we provide a comprehensive summary of the performance of the 36 different tables from 3 to 38 that are individually presented for each dataset across different model selection methods in the subsection A.1.1. The initial summary, depicted in Table 39, illustrates the average accuracy per dataset. This summary considers accuracy average of all classifiers evaluated using the three model selection methods: Source Risk, IWCV, and Target Risk. The second summary, displayed in Table 40, showcases the average accuracy per classifier across all datasets, again evaluated using the three model selection methods.

From both tables 39 and 39, it is evident that the Target Risk selection method emerges as the most effective for hyperparameter selection. This is expected as it utilizes the target labels, making it akin to an upper bound. The difference between the IWCV and Source Risk methods is not substantial, indicating similar performances.

Table 39 highlights the difficulties or the challenges of the different datasets. Notably, the most challenging ones include OnHWeq, wisdm, sports activities, and sleep stages. Among these, two datasets, Wisdm 51 and sleep stages 48, stand out as particularly imbalanced, further complicating the classification task. OnHWeq also presents difficulty due to its high number of classes (15 classes).

Table 40 reaffirms the observations and conclusions derived from the critical diagrams presented in sections 5.1 and A.2.1 concerning the ranking of algorithms and the performance of the backbones.

Table 39: Accuracy results summary table of average accuracy per dataset valuated the 3 model selection methods: Source Risk, IWCV and Target Risk

| dataets | IWCV | Source Risk | Target Risk |
|---|---|---|---|
| cwrBearing | $0.79 \pm 0.02$ | $0.81 \pm 0.02$ | $0.83 \pm 0.04$ |
| ford | $0.56 \pm 0.05$ | $0.76 \pm 0.02$ | $0.77 \pm 0.02$ |
| har | $0.76 \pm 0.06$ | $0.71 \pm 0.04$ | $0.82 \pm 0.04$ |
| hhar | $0.62 \pm 0.04$ | $0.62 \pm 0.03$ | $0.71 \pm 0.03$ |
| mfd | $0.63 \pm 0.09$ | $0.63 \pm 0.09$ | $0.71 \pm 0.07$ |
| TimeMatch | $0.63 \pm 0.06$ | $0.64 \pm 0.03$ | $0.68 \pm 0.04$ |
| OnHWeq | $0.49 \pm 0.02$ | $0.49 \pm 0.02$ | $0.54 \pm 0.02$ |
| ptbXLecg | $0.6 \pm 0.02$ | $0.61 \pm 0.02$ | $0.62 \pm 0.02$ |
| sleepStage | $0.62 \pm 0.03$ | $0.64 \pm 0.04$ | $0.7 \pm 0.02$ |
| sportsActivities | $0.52 \pm 0.05$ | $0.51 \pm 0.07$ | $0.61 \pm 0.07$ |
| Muscle | $0.55 \pm 0.03$ | $0.53 \pm 0.04$ | $0.64 \pm 0.05$ |
| wisdm | $0.51 \pm 0.06$ | $0.45 \pm 0.07$ | $0.59 \pm 0.06$ |

Table 40: Accuracy results summary table evaluated the 3 model selection methods: Source Risk, IWCV and Target Risk

| classifier | IWCV | Source Risk | Target Risk |
|---|---|---|---|
| Source only | $0.59 \pm 0.07$ | $0.6 \pm 0.06$ | $0.7 \pm 0.05$ |
| CoDATS | $0.64 \pm 0.07$ | $0.66 \pm 0.05$ | $0.76 \pm 0.04$ |
| CoTMix | $0.57 \pm 0.05$ | $0.57 \pm 0.04$ | $0.66 \pm 0.03$ |
| InceptionCDAN | $0.64 \pm 0.05$ | $0.63 \pm 0.04$ | $0.75 \pm 0.06$ |
| InceptionDANN | $0.63 \pm 0.06$ | $0.67 \pm 0.06$ | $0.75 \pm 0.05$ |
| InceptionMix | $0.56 \pm 0.07$ | $0.57 \pm 0.1$ | $0.63 \pm 0.08$ |
| InceptionRain | $0.67 \pm 0.06$ | $0.68 \pm 0.07$ | $0.77 \pm 0.04$ |
| OTDA | $0.48 \pm 0.06$ | $0.5 \pm 0.04$ | $0.53 \pm 0.03$ |
| Raincoat | $0.66 \pm 0.06$ | $0.65 \pm 0.06$ | $0.76 \pm 0.05$ |
| SASA | $0.45 \pm 0.03$ | $0.48 \pm 0.02$ | $0.5 \pm 0.03$ |
| VRADA | $0.5 \pm 0.05$ | $0.5 \pm 0.05$ | $0.52 \pm 0.06$ |
| Target only | $0.9 \pm 0.04$ | $0.9 \pm 0.04$ | $0.9 \pm 0.04$ |

### A.1.3 DATA IMBALANCE ANALYSIS

In this section we propose to quantify the data imbalance in both source (train, test) and target (train, test). Following what is done in Olson et al. (2017), the degree of class imbalance in each dataset is measured using a score $I \in [0, 1)$, where 0 represents perfectly balanced classes, and values approaching 1 indicate extreme class imbalance. $I$ score is computed as follow:

$$I = k \sum_{i=1}^{k} (\frac{n_i}{N} - \frac{1}{k})^2$$

where $k$ is the number of classes, $n_i$ is the number of instances for class $i$, and $N$ is the total number of samples. This calculation captures the deviation of class distribution from perfect balance, assigning higher values as the imbalance increases, particularly when a single class dominates the dataset. Tables from 41 to 51 present the used samples and the $I$ score for both source (train, test) and target (train, test). We define a threshold, categorizing datasets as highly imbalanced if their I scores is higher than 0.6. We therefore note than that only 3 datasets out of 12 presents imbalanced class ratios: ptbXLecg dataset, sleepStage dataset and wisdm dataset. Finally we should note that out of these three imabalanced datasets, ptbXLecg is the sole dataset proposed in this benchmark.

Table 41: Data Imbalance stats on Ford dataset

| | | Source | | | | Target | | | |
|---|---|---|---|---|---|---|---|---|---|
| | K | N_train | N_test | I_train | I_test | N_train | N_test | I_train | I_test |
| Ford_A-Ford_B | 2 | 3601 | 1320 | 0.001 | 0.001 | 648 | 162 | 0 | 0 |
| Ford_B-Ford_A | 2 | 648 | 162 | 0 | 0 | 3601 | 1320 | 0.001 | 0.001 |

Table 42: Data Imbalance stats on cwrBearing dataset

| | | Source | | | | Target | | | |
|---|---|---|---|---|---|---|---|---|---|
| | K | N_train | N_test | I_train | I_test | N_train | N_test | I_train | I_test |
| 0_1 | 4 | 7511 | 1877 | 0.284 | 0.348 | 10081 | 2520 | 0.208 | 0.292 |
| 0_3 | 4 | 7511 | 1877 | 0.284 | 0.348 | 10256 | 2563 | 0.207 | 0.304 |
| 1_3 | 4 | 10081 | 2520 | 0.208 | 0.292 | 10256 | 2563 | 0.207 | 0.304 |
| 2_0 | 4 | 10260 | 2565 | 0.207 | 0.305 | 7511 | 1877 | 0.284 | 0.348 |
| 3_0 | 4 | 10256 | 2563 | 0.207 | 0.304 | 7511 | 1877 | 0.284 | 0.348 |

## A.2 FURTHER ANALYSIS OF THE BENCHMARK

### A.2.1 AVERAGE RANKS ON ALL EXPERIMENTS

The average accuracy rank of all algorithms and model selection methods is illustrated in Figure 6. Different colors are used to distinguish hyperparameter tuning methods for model selection. IWCV

Table 43: Data Imbalance stats on har dataset

|  | K | Source | | | | Target | | | |
| --- | --- | --- | --- | --- | --- | --- | --- | --- | --- |
|  |  | N_train | N_test | I_train | I_test | N_train | N_test | I_train | I_test |
| 2_11 | 6 | 211 | 91 | 0.008 | 0.009 | 221 | 95 | 0.008 | 0.009 |
| 12_16 | 6 | 224 | 96 | 0.011 | 0.009 | 256 | 110 | 0.035 | 0.043 |
| 9_18 | 6 | 201 | 87 | 0.005 | 0.006 | 254 | 110 | 0.011 | 0.012 |
| 6_23 | 6 | 227 | 98 | 0.005 | 0.003 | 260 | 112 | 0.015 | 0.019 |
| 7_13 | 6 | 215 | 93 | 0.005 | 0.004 | 228 | 99 | 0.009 | 0.009 |

Table 44: Data Imbalance stats on hhar dataset

|  | K | Source | | | | Target | | | |
| --- | --- | --- | --- | --- | --- | --- | --- | --- | --- |
|  |  | N_train | N_test | I_train | I_test | N_train | N_test | I_train | I_test |
| a_b | 6 | 2900 | 746 | 0.007 | 0.008 | 3227 | 831 | 0.012 | 0.011 |
| c_d | 6 | 3027 | 785 | 0.002 | 0.002 | 2863 | 738 | 0.009 | 0.009 |
| e_f | 6 | 3120 | 805 | 0.005 | 0.005 | 2821 | 726 | 0.004 | 0.004 |
| g_h | 6 | 3169 | 818 | 0.012 | 0.012 | 3111 | 806 | 0.004 | 0.004 |
| i_a | 6 | 3264 | 839 | 0.001 | 0.001 | 2900 | 746 | 0.007 | 0.008 |

Table 45: Data Imbalance stats on mfd dataset

|  | K | Source | | | | Target | | | |
| --- | --- | --- | --- | --- | --- | --- | --- | --- | --- |
|  |  | N_train | N_test | I_train | I_test | N_train | N_test | I_train | I_test |
| 0_1 | 3 | 1828 | 901 | 0.263 | 0.264 | 1828 | 901 | 0.263 | 0.264 |
| 0_3 | 3 | 1828 | 901 | 0.263 | 0.264 | 1828 | 901 | 0.263 | 0.264 |
| 1_0 | 3 | 1828 | 901 | 0.263 | 0.264 | 1828 | 901 | 0.263 | 0.264 |
| 1_2 | 3 | 1828 | 901 | 0.263 | 0.264 | 1828 | 901 | 0.263 | 0.264 |
| 1_3 | 3 | 1828 | 901 | 0.263 | 0.264 | 1828 | 901 | 0.263 | 0.264 |

Table 46: Data Imbalance stats on OnHWeq dataset

|  | K | Source | | | | Target | | | |
| --- | --- | --- | --- | --- | --- | --- | --- | --- | --- |
|  |  | N_train | N_test | I_train | I_test | N_train | N_test | I_train | I_test |
| R_L | 15 | 27750 | 11893 | 0.087 | 0.087 | 4396 | 1884 | 0.097 | 0.098 |
| L_R | 15 | 4396 | 1884 | 0.097 | 0.098 | 27750 | 11893 | 0.087 | 0.087 |

Table 47: Data Imbalance stats on ptbXLecg dataset

|  | K | Source | | | | Target | | | |
| --- | --- | --- | --- | --- | --- | --- | --- | --- | --- |
|  |  | N_train | N_test | I_train | I_test | N_train | N_test | I_train | I_test |
| 0_1 | 5 | 5424 | 1356 | **0.933** | **0.933** | 3708 | 927 | **0.961** | **0.965** |
| 0_2 | 5 | 5424 | 1356 | **0.933** | **0.933** | 3055 | 764 | **0.679** | **0.676** |
| 0_3 | 5 | 5424 | 1356 | **0.933** | **0.933** | 307 | 77 | 0.383 | 0.387 |
| 1_3 | 5 | 3708 | 927 | **0.961** | **0.965** | 307 | 77 | 0.383 | 0.387 |
| 2_1 | 5 | 3055 | 764 | **0.679** | **0.676** | 3708 | 927 | **0.961** | **0.965** |

Table 48: Data Imbalance stats on sleepStage dataset

|  | K | Source | | | | Target | | | |
| --- | --- | --- | --- | --- | --- | --- | --- | --- | --- |
|  |  | N_train | N_test | I_train | I_test | N_train | N_test | I_train | I_test |
| 16_1 | 5 | 1502 | 645 | 0.39 | 0.388 | 1602 | 687 | **0.713** | **0.715** |
| 9_14 | 5 | 1565 | 672 | 0.567 | 0.571 | 1369 | 587 | 0.372 | 0.37 |
| 12_5 | 5 | 1420 | 609 | 0.278 | 0.277 | 1342 | 576 | 0.397 | 0.396 |
| 7_18 | 5 | 1574 | 675 | 0.211 | 0.212 | 1318 | 566 | 0.247 | 0.248 |
| 0_11 | 5 | 1377 | 591 | 0.191 | 0.191 | 1211 | 519 | **0.711** | **0.709** |

Table 49: Data Imbalance stats on sportsActivities dataset

| | K | Source | | | | Target | | | |
| | | N_train | N_test | I_train | I_test | N_train | N_test | I_train | I_test |
|---|---|---|---|---|---|---|---|---|---|
| p1_p8 | 19 | 912 | 228 | 0 | 0 | 912 | 228 | 0 | 0 |
| p7_p3 | 19 | 912 | 228 | 0 | 0 | 912 | 228 | 0 | 0 |
| p4_p2 | 19 | 912 | 228 | 0 | 0 | 912 | 228 | 0 | 0 |
| p5_p6 | 19 | 912 | 228 | 0 | 0 | 912 | 228 | 0 | 0 |
| p5_p4 | 19 | 912 | 228 | 0 | 0 | 912 | 228 | 0 | 0 |

Table 50: Data Imbalance stats on UltrasoundMuscleContraction dataset

| | K | Source | | | | Target | | | |
| | | N_train | N_test | I_train | I_test | N_train | N_test | I_train | I_test |
|---|---|---|---|---|---|---|---|---|---|
| sb1_sb8 | 2 | 73600 | 18400 | 0.162 | 0.162 | 32000 | 8000 | 0.026 | 0.026 |
| sb8_sb6 | 2 | 32000 | 8000 | 0.026 | 0.026 | 8000 | 2000 | 0.084 | 0.084 |
| sb2_sb7 | 2 | 1600 | 400 | 0.003 | 0.003 | 16000 | 4000 | 0.006 | 0.006 |
| sb5_sb4 | 2 | 8000 | 2000 | 0.08 | 0.08 | 16000 | 4000 | 0 | 0 |
| sb3_sb5 | 2 | 15097 | 3775 | 0.001 | 0.002 | 8000 | 2000 | 0.08 | 0.08 |

Table 51: Data Imbalance stats on wisdm dataset

| | K | Source | | | | Target | | | |
| | | N_train | N_test | I_train | I_test | N_train | N_test | I_train | I_test |
|---|---|---|---|---|---|---|---|---|---|
| 5_1 | 6 | 174 | 46 | **0.653** | 0.531 | 184 | 48 | **0.736** | **0.682** |
| 3_5 | 6 | 214 | 58 | 0.598 | 0.534 | 174 | 46 | 0.653 | 0.531 |
| 27_3 | 6 | 214 | 57 | **0.653** | 0.564 | 214 | 58 | 0.598 | 0.534 |
| 21_31 | 6 | 215 | 58 | 0.453 | 0.377 | 271 | 71 | **0.646** | 0.601 |
| 7_30 | 6 | 188 | 51 | 0.5 | 0.46 | 156 | 41 | **0.751** | **0.686** |

method is represented by the green color, the Source Risk method by blue, and the Target Risk method by red. As expected, the initial top cluster encompasses different algorithms when employing the Target Risk model selection method. This is because the target labels are utilized to choose the optimal hyperparameters, which is not feasible in the unsupervised domain adaptation setup. The IWCV and Source Risk methods, as previously discussed in Section 5.2, have quite similar performances. As depicted in the figure 6, it is challenging to discern distinct clusters for either method (blue and green are visually evenly distributed after the top red cluster). This is explained by the fact that the Source Risk method is robust in the presence of small domain shifts. However, as the shift magnitude escalates, the robustness of the Source Risk method diminishes. Consequently, the utilization of an alternative method, such as IWCV, becomes more advantageous for effectively adapting to substantial shifts. In our benchmarking experiments, where diverse datasets and distinct pairs are examined, we observe varying degrees of shifts — small, medium, or large — between the source and target domains. This variability contributes to the convergence in performance between the two methods (Source Risk and IWCV) within the scope of this paper. For both the IWCV and Source Risk methods, InceptionRain consistently exhibits the highest average rank among all nine models. As previously discussed in Section 5.2, we observe a slight non significant decrease in InceptionRain's performance compared to other classifiers (mainly to InceptionDANN) when employing Target Risk for hyperparameter selection. Additionally, three main discernible clusters of algorithms are visible in this diagram. The first cluster, comprising VRADA and OTDA, represents the two poorest classifiers across all three model selection methods. The second cluster encompasses all algorithms utilizing both IWCV and Source Risk methods, alongside Inception-Mix, CoTMix, and Inception with the Target Risk method, demonstrating very similar or closely ranked performance. The third cluster, featuring InceptionDANN, InceptionRain, InceptionCDAN, RainCoat, and CoDATS using the Target Risk method, showcases the superior performance of these algorithms among all tested methods.

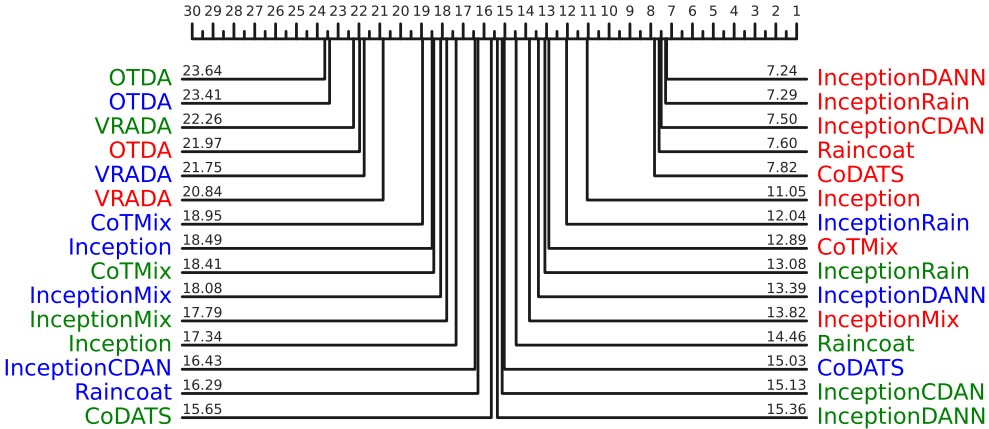

Figure 6: Average rank diagrams, using accuracy on all tested algorithms and hyperparameter tuning methods. The different colors specify the employed hyperparameter tuning method: green for IWCV IWCV, blue for Source Risk and red for Target Risk

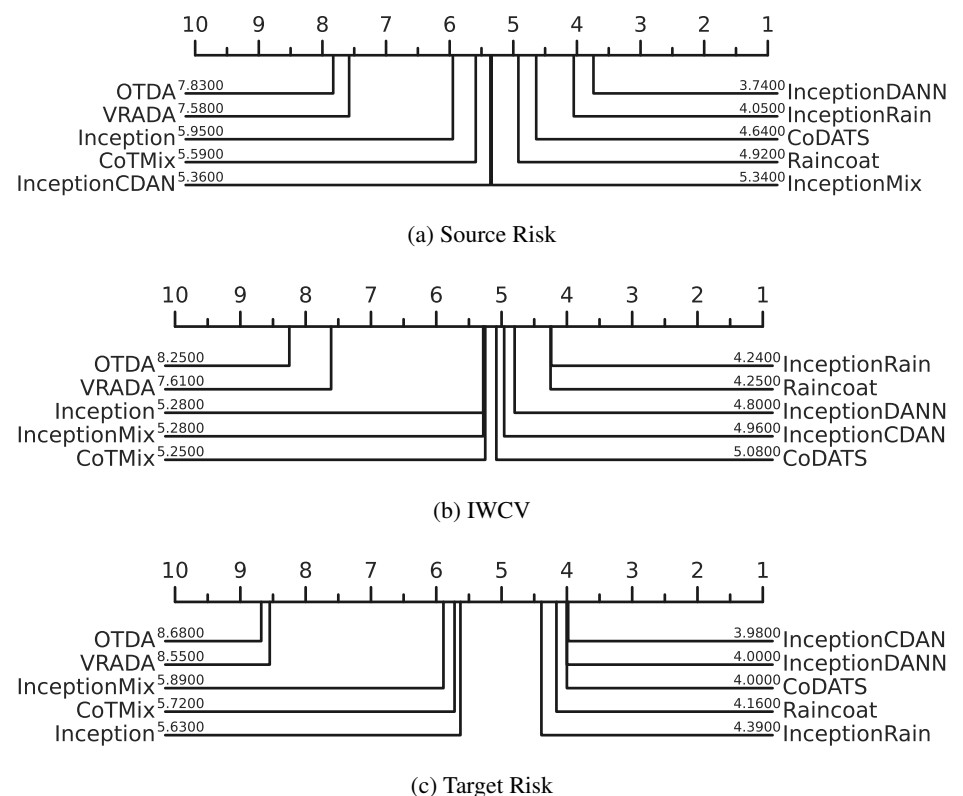

Figure 7: Average rank diagrams, using F1-score, with different criteria for hyperparameter tuning

### A.2.2 AVERAGE RANKS WITH F1-SCORE

Figure 7 showcases the average rank of all algorithms based on F1-score and depending on the hyperparameter tuning method. These diagrams are very similar to the ones for accuracy, except for a few inversions of some algorithms such as InceptionRain and InceptionDANN on Figure 7a or CoTMix and Inception on Figure 7b, which have very similar average ranks. There are more

differences on Figure 7c, but again these differences concern algorithms of similar rank. Overall, our conclusion hold, therefore the accuracy is a sufficient metric to evaluate the performance of the models.

### A.2.3 Accuracy of algorithms over all datasents

The distribution of the predicted accuracy of the target test by different classifier using the three methods of model selection (IWCV 8a, source risk 8c and target risk 8e) can be visually displayed by the violin plots in 8. The median is marked with a dashed line, and the range of quartile is represented by a dotted line. The more centralized the data distribution is, the wider the violin is. As concluded before in section 5.2, we can see again in these plots that InceptionRain displays the best performance compared to other classifiers, in terms of median and quartiles of the accuracy. IWCV presents more variance than Source Risk, we are convinced that estimating time series data with only 5 Gaussian is not enough. Source Risk is a good method for hyprameter tuning if the shift between the distribution is not very big.

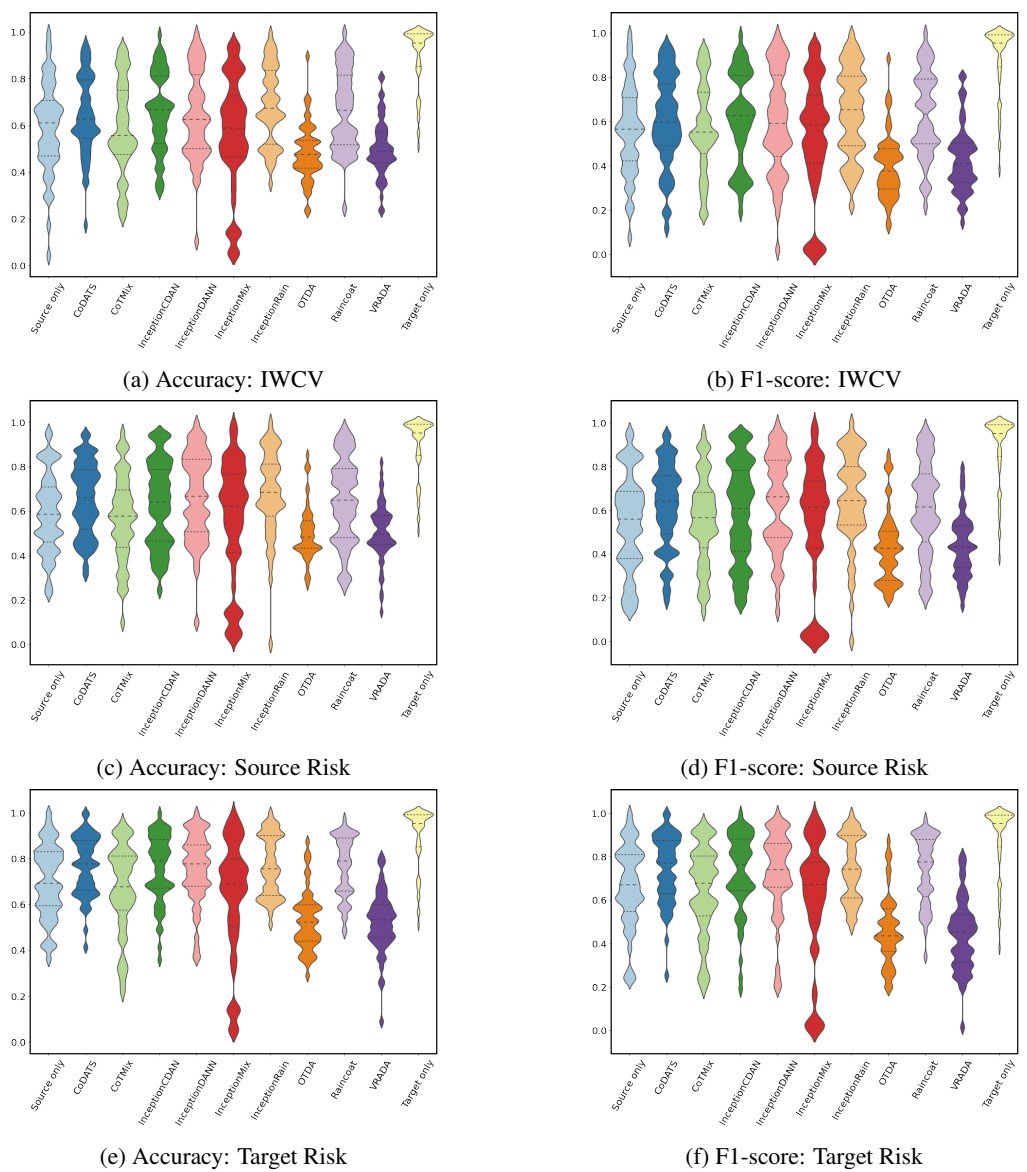

(a) Accuracy: IWCV             (b) F1-score: IWCV

(c) Accuracy: Source Risk         (d) F1-score: Source Risk

(e) Accuracy: Target Risk         (f) F1-score: Target Risk

Figure 8: Violin plots of algorithms performance, accuracy on the left and F1-score on the right, per hyperparameter selection method for all the run experiments

### A.2.4 FURTHER ANALYSIS FOR IMBALANCED DATASETS

Since we noticed some imbalanced datasets in our experiments we present additionally the violin plots with the F1-score metric. Figure 8 presents the distribution of the predicted F1-score of the target test by different classifier using the three methods of model selection (IWCV 8b, Source Risk 8d and Target Risk 8f) can be visually displayed by the violin plots in 8.

### A.2.5 ANALYSIS OF HYPERPARAMETER TUNING APPROACHES

Figure 9 displays several examples of hyperparameter tuning for different datasets/pairs and for different UDA algorithms. The graphs from this figure show the values of each tuning approach for each set of hyperparameters, where the indices are sorted with respect to the target accuracy on the validation set. We can see that there exists a correlation between the target risk and the two proxy used, as the source risk and IWCV both show a tendency to evolve similarly as the target risk. However, we can note on one hand that the Source Risk displays a plateau behaviour, with many hyperparameter sets having similar values, while their target risk is clearly different. It thus makes it difficult for the source risk to differentiate between models the ones more probable to get good target performance. On the other hand, IWCV sometimes shows better values (lower cross entropy) for hyperparameters having higher target accuracy, as it is the case on Fig. 9b, but at other times it selects a hyperparameter set at much lower target accuracy than the best possible, for instance for Figs 9a and 9d. These graphs thus explain the gap of performance between target risk on one side, and source risk and IWCV on the other side.

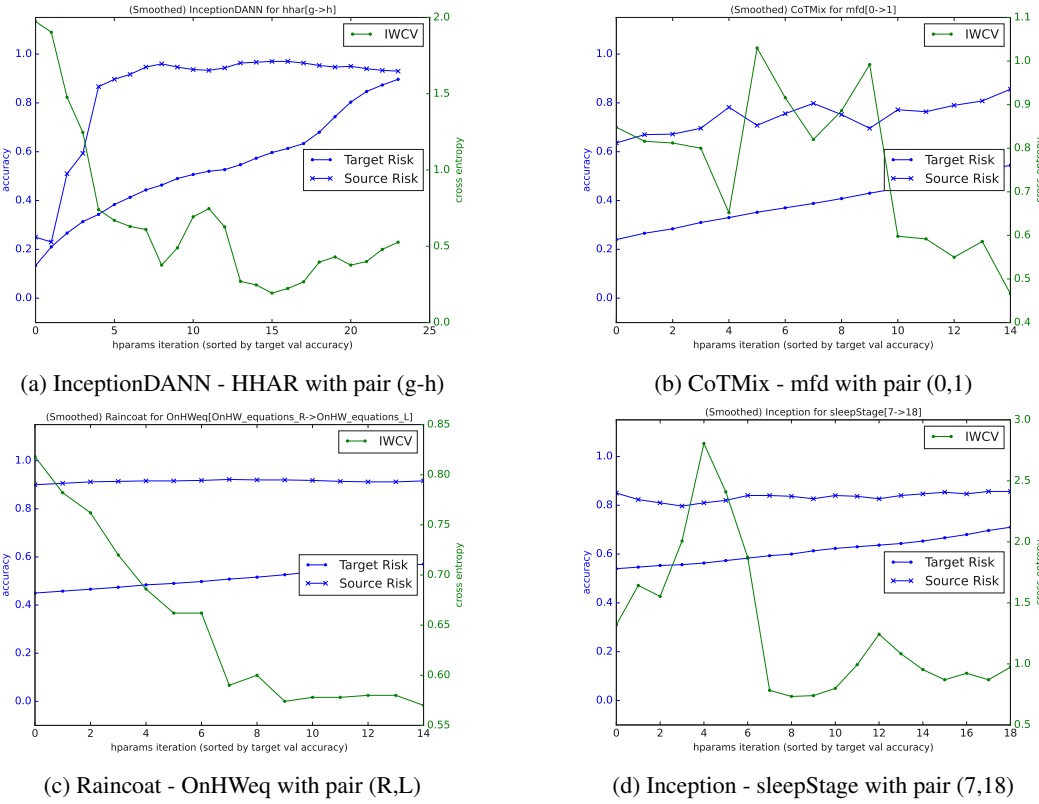

(a) InceptionDANN - HHAR with pair (g-h)      (b) CoTMix - mfd with pair (0,1)

(c) Raincoat - OnHWeq with pair (R,L)      (d) Inception - sleepStage with pair (7,18)

Figure 9: Accuracy and cross entropy loss during the hyperparameter tuning stage for several datasets and UDA algorithms.

Figure 10 compares the behaviour of the hyperparameter tuning approaches for several algorithms on the HAR dataset for the same pair of source-target. The previous analysis is still valid here, which suggests that it does not depend much on the UDA algorithm but probably more on the dataset.

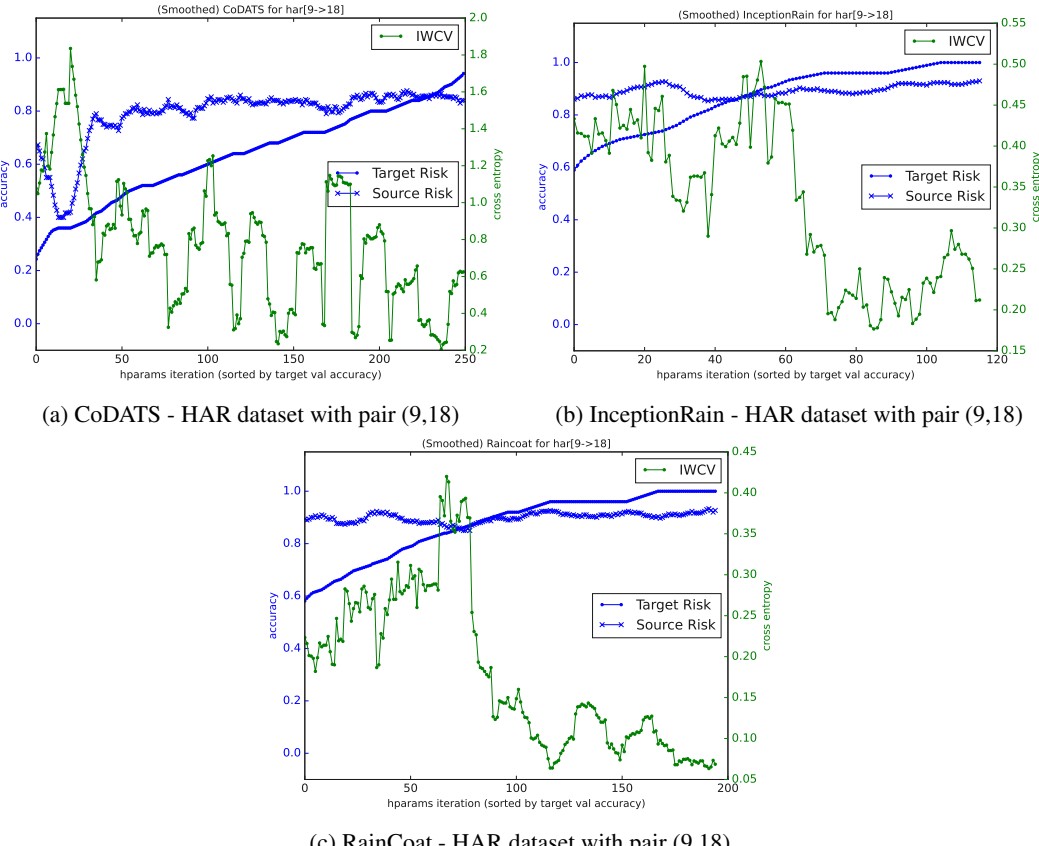

(a) CoDATS - HAR dataset with pair (9,18)    (b) InceptionRain - HAR dataset with pair (9,18)

(c) RainCoat - HAR dataset with pair (9,18)

Figure 10: Accuracy and cross entropy loss during the hyperparameter tuning stage for HAR dataset and 3 UDA algorithms.

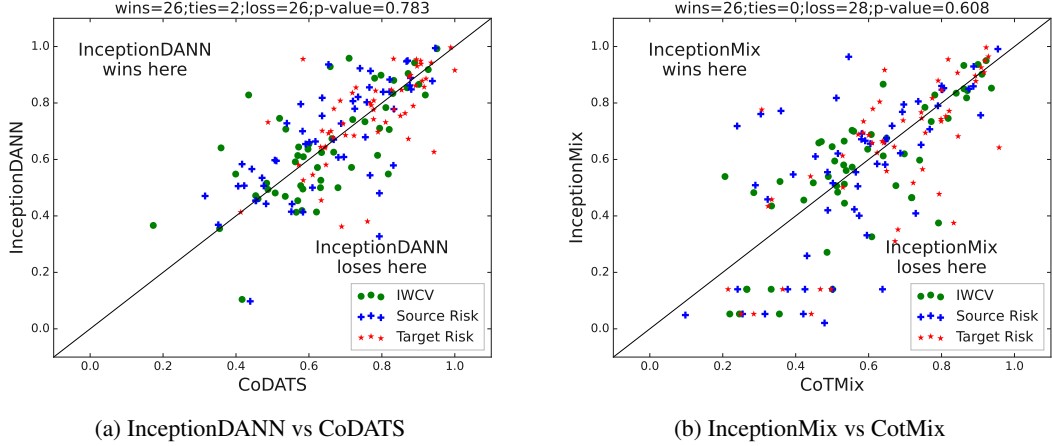

(a) InceptionDANN vs CoDATS    (b) InceptionMix vs CotMix

Figure 11: Pairwise accuracy comparison between CoDATS, CotMix and Raincoat against the same methods with Inception backbone instead of the backbone originally proposed. The statistical analysis and comparison above the figures are computed using only results based on IWCV.

### A.2.6    DETAILED COMPARISON OF BACKBONES

This section gives additional details about the importance of backbones. Figures 11 compare the performance of the three original backbones used in CoDATS, CotMix and Raincoat against an

Inception backbone. Consistent with the averaged result presented in Figure 5, there is no significant differences between the 3 original backbones and the Inception backbone.

Figures 12 provides a deeper analysis on the four different themes of dataset. Firstly, as the number of data points are sometimes quite small, the confidence on the statistical analysis is small. Therefore, there is no statistical difference in those graphs. There is still two interesting observations, firstly the original backbones get the best performance compared to Inception in datasets related to motion. This could be explained by the fact that most of the established TSC UDA dataset (three out of five in total) are part of the motion theme, thus leading to a bias that those models where designed to work for those types of datasets. Secondly, there is a couple of datasets (miniTimeMatch and sportsActivity), where InceptionMix always predict the same classes thus has a very low accuracy. As this is also true for the Target Risk, the reason does not seem to be the hyperparameter search but rather an unexpected mismatch between those two specifics datasets, CoTMix algorithm and the Inception backbone.

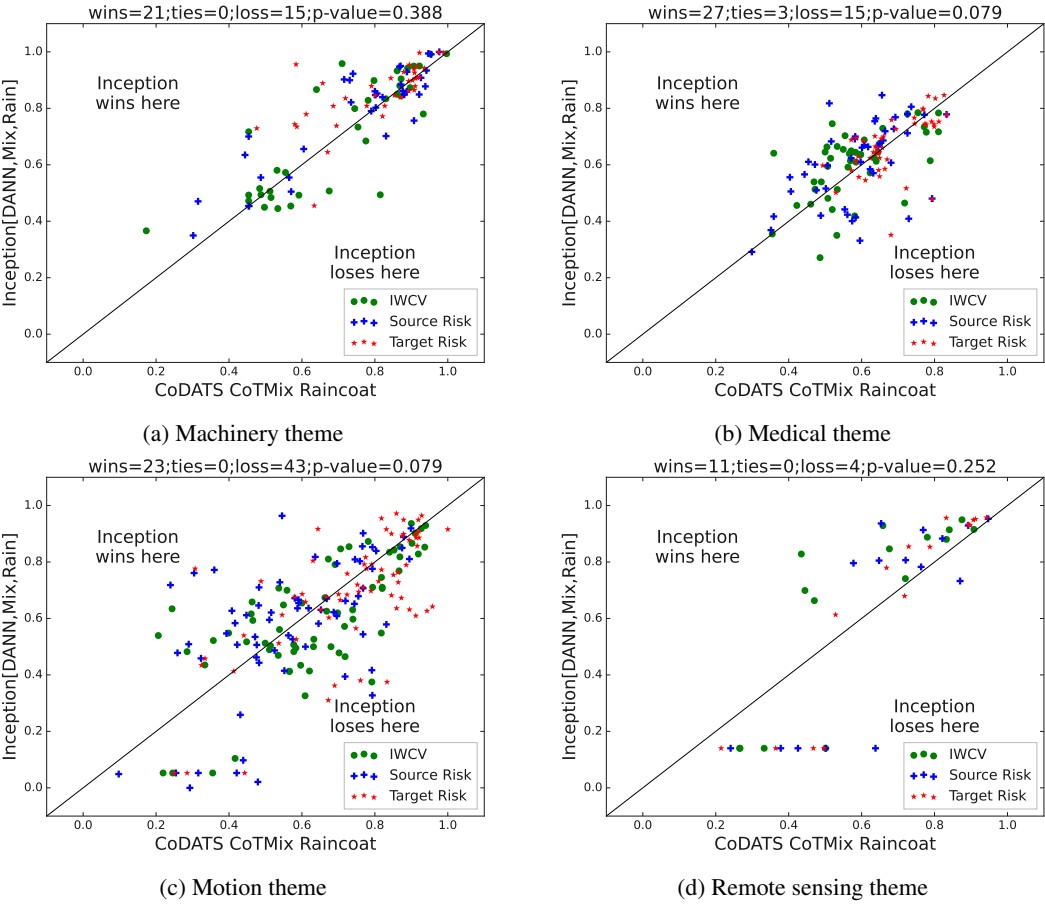

Figure 12: Pairwise accuracy comparison between original backbone against Inception backbone for the 4 different dataset themes. The statistical analysis and comparison above the figures are computed using only results based on IWCV.

### A.2.7 META-FEATURES INTEROPERABILITY OF THE BENCHMARK

Within this section, our aim is to offer a deeper and high level analysis of the various algorithms' performance and experiments showcased in the benchmark, employing explainability and interpretability techniques facilitated by metadata. The metadata, or dataset characteristics, we consider here consists of the following quantities. "Shift proxy" denotes a proxy of the data shift computed as the relative difference of Inception's performance (i.e. with no DA) between target and source. "Pair std" denotes the variability of Inception's performance (with no DA) among the different pairs

of source-data, which can be considered as another proxy of the shift. "Imbalance" is the imbalance score of the data as computed in A.1.3. Finally, "No. classes", "No. instances" and "Length" respectively denote the number of classes, the number of time series in the training set, and the length of the time series, as provided in Table 2.

In order to assess how these metadata affect the performance of each UDA algorithm, we propose to run a regression task with the metadata as input and the considered algorithm's performance as output, using XGBoost. We then extract the feature importance scores provided by XGBoost. Figure 13 display the importance score of the metadata for each UDA algorithm, and Figures 14 to 22 show the evolution of each algorithm's performance with respect to its 4 most important metadata. The latter plots are made with seaborn's regplot, which also provide an estimation of the relationship and its variance (the colors indicate the metadata considered). From these plots, we can easily see that the UDA algorithms are affected differently by the metadata. The shift proxy seems to affect most of the methods, whose performance decrease as the shift increases. It is particularly true for InceptionCDAN and InceptionRain. Note however that it is expected as those methods rely on the Inception backbone, which we used for the estimation of the shift. The number of classes affect mostly CoDATS, VRADA, InceptionMix, and to a lesser extend InceptionCDAN and InceptionDANN. Indeed, performance seems to be decreasing with the number of classes. The length of the time series is found to be important for OTDA, Raincoat, and InceptionDANN. This is understandable for OTDA, as the time series are flattened. Concerning Raincoat, the reason stems from the architecture of the initial step of time series alignment, where the proposed CNN backbone relies on a fixed kernel size. However, it is observed that the significance of time series length diminishes when using the Inception backbone. The Inception backbone, by design, is not negatively impacted by the length, as it incorporates different kernel sizes in parallel, facilitating adaptation to the length of the time series. The variability among pairs mostly affect CoTMix, InceptionCDAN and InceptionRain, who seem to have better performance when the pairs look less alike. Finally, the imbalance in data mostly affect InceptionRain, while the number of instances seems to have an impact only for Raincoat and InceptionRain.

## A.3 DETAILS OF LEARNING ALGORITHMS

Let $\mathcal{X}$ be the input space. Let $\mathcal{Y}$ be the label space, a vector space where the labels represented as one-hot vectors and the predictions represented as categorical probabilities live. $D_S$ is the source domain consisting of $n_S$ labeled time series. For a sample $(\mathbf{X}^s, \mathbf{y}^s) \in D_S$ let $\mathbf{X}^s \in \mathcal{X}$ represent the time series data and $\mathbf{y}^s \in \mathcal{Y}$ represent the corresponding label. $D_T \subset \mathcal{X}$ is the target domain consisting solely of $n_T$ unlabeled time series. $\mathcal{Z}$ is the latent space.

**Time series classification:** Let $E : \mathcal{X} \to \mathcal{Z}$ be the backbone. The backbone is usually a highly non-linear function, such as a neural network, designed according to the input data. In the case of time series data, the backbone can be a RNN, a 1D CNN. Its role is to encode the input into the latent space. Let $C : \mathcal{Z} \to \mathcal{Y}$ be the classifier, mapping the latent space representation of the input into a categorical probability.

Let $\mathcal{L} : \mathcal{Y} \times \mathcal{Y} \to \mathbb{R}$ be a loss function, such as the cross-entropy, which evaluates the quality of the classifier by assigning a number to each label-prediction pair. Let $D_S = \{(\mathbf{X}_i^s, \mathbf{y}_i^s)\}_{i=1}^{n_S}$ be the source domain. In the traditional time series classification framework, the classification loss is,

$$\mathcal{L}_C(E, C) = \frac{1}{n_S} \sum_{i=1}^{n_S} \mathcal{L}\left(C\left(E(\mathbf{X}_i^s)\right), \mathbf{y}_i^s\right) . \tag{1}$$

The learning objective is to minimize this loss with respect to the classifier $C$ and the backbone $E$.

**Adversarial domain adaptation:** The goal of this learning framework is to achieve domain adaptation by forcing the backbone to produce a latent representation that is domain invariant. A discriminator $D : \mathcal{Z} \to \{\mathbf{y}_{\text{domain}}^s, \mathbf{y}_{\text{domain}}^t\}$ is introduced with the task of separating source and target domain samples, where $\mathbf{y}_{\text{domain}}^s$ and $\mathbf{y}_{\text{domain}}^t$ are binary labels indicating that the corresponding sample belongs to the source or target domain, respectively. During training the discriminator learns how to separate source domain samples from target domain samples and the backbone learns how to fool the discriminator.

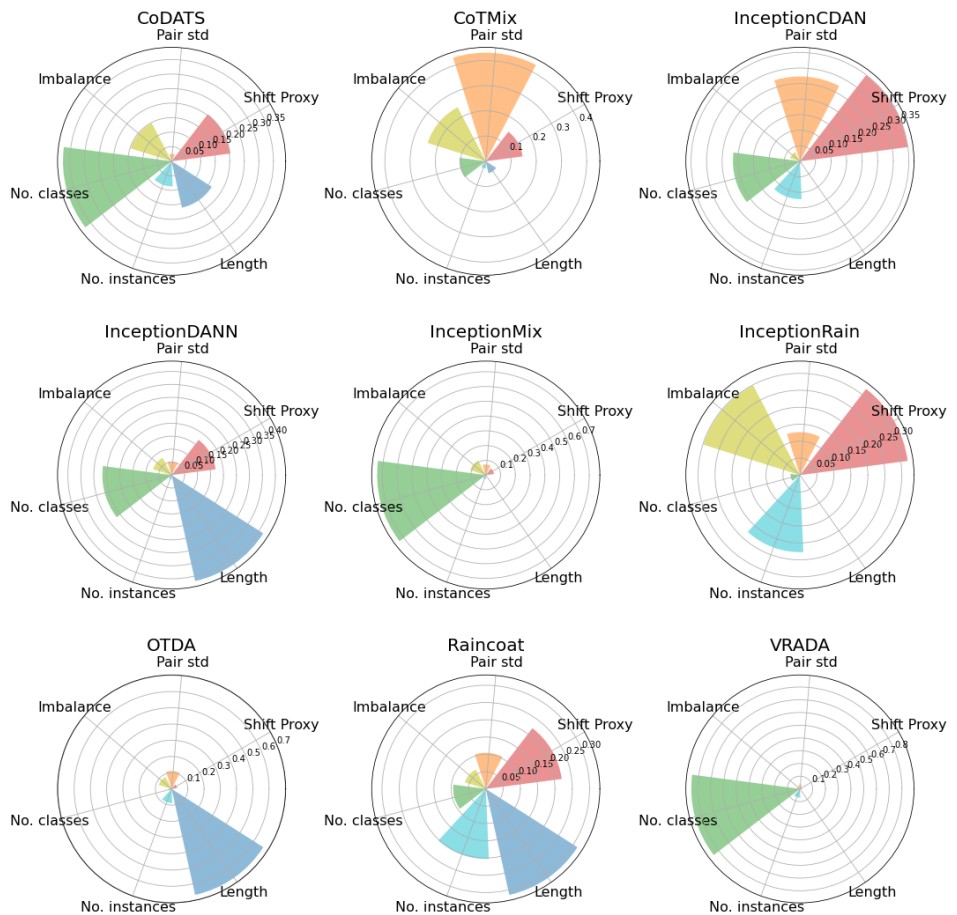

Figure 13: Impact of different dataset characteristics (metadata) on the performance of UDA algorithms, measured from XGBoost's feature importance score.

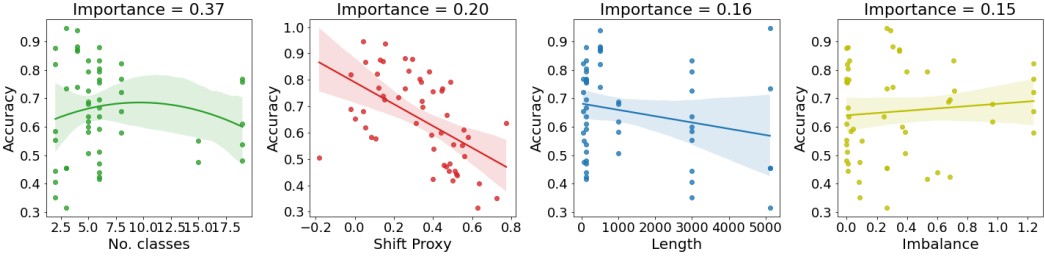

Figure 14: Visualization of CoDATS performance versus its four more important data characteristics returned from XGBoost.

Let $D_S = \{(\mathbf{X}_i^s, \mathbf{y}_i^s)\}_{i=1}^{n_S}$ and $D_T = \{\mathbf{X}_i^t\}_{i=1}^{n_T}$ be the source and target domains, respectively. The adversarial loss is,

$$\mathcal{L}_A(E, D) = \frac{1}{n_S} \sum_{i=1}^{n_S} \mathcal{L}\left(D\left(E(\mathbf{X}^s)\right), \mathbf{y}_{\text{domain}}^s\right) + \frac{1}{n_T} \sum_{i=1}^{n_T} \mathcal{L}\left(D\left(E(\mathbf{X}^t)\right), \mathbf{y}_{\text{domain}}^t\right), \quad (2)$$

where $\mathcal{L}$ is a binary loss function, such as the binary cross entropy loss. The total loss for the adversarial domain adaptation framework is,

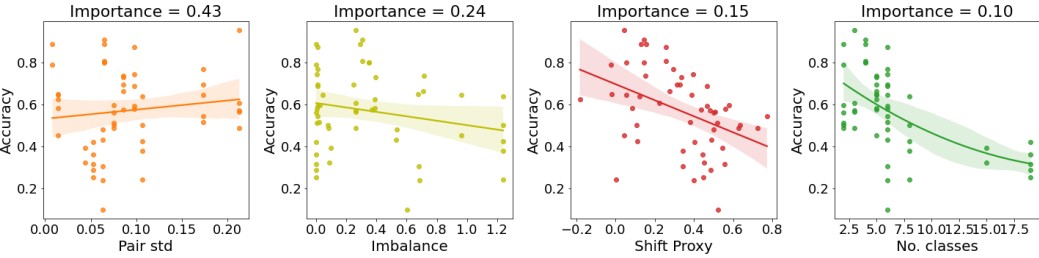

Figure 15: Visualization of CoTMix performance versus its four more important data characteristics returned from XGBoost.

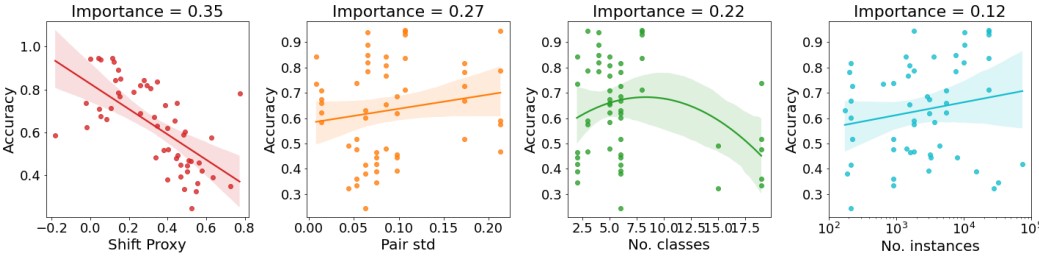

Figure 16: Visualization of InceptionCDAN performance versus its four more important data characteristics returned from XGBoost.

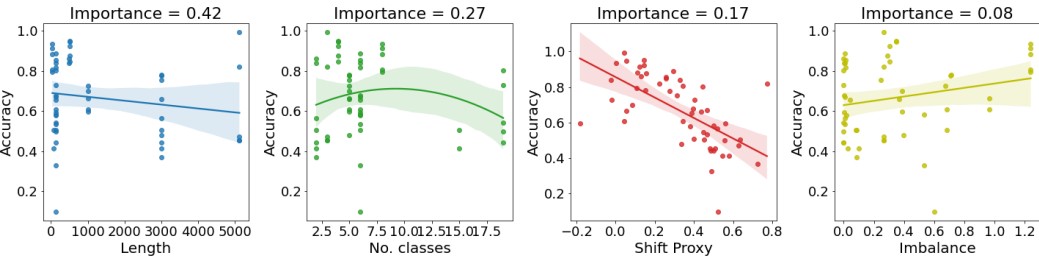

Figure 17: Visualization of InceptionDANN performance versus its four more important data characteristics returned from XGBoost.

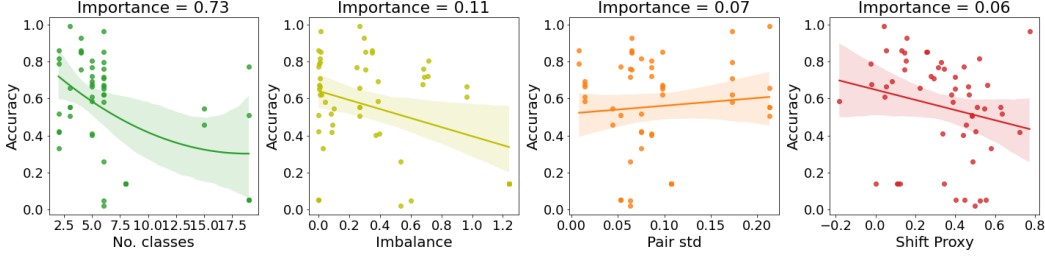

Figure 18: Visualization of InceptionMix performance versus its four more important data characteristics returned from XGBoost.

$$\mathcal{L}_{\text{total}}(E, C, D) = \mathcal{L}_C(E, C) - \lambda \mathcal{L}_A(E, D) , \tag{3}$$

Where $\lambda$ is a trade-off between the adversarial and classification losses. During training, the objective 3 is minimized with respect to the classifier $C$ and encoder $E$ and maximized with respect to the discriminator $D$.

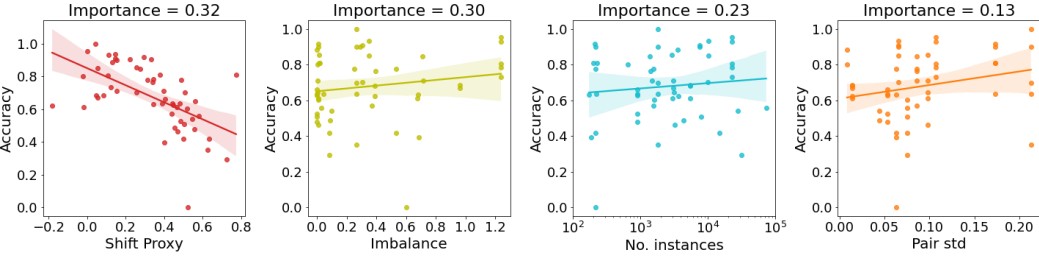

Figure 19: Visualization of InceptionRain performance versus its four more important data characteristics returned from XGBoost.

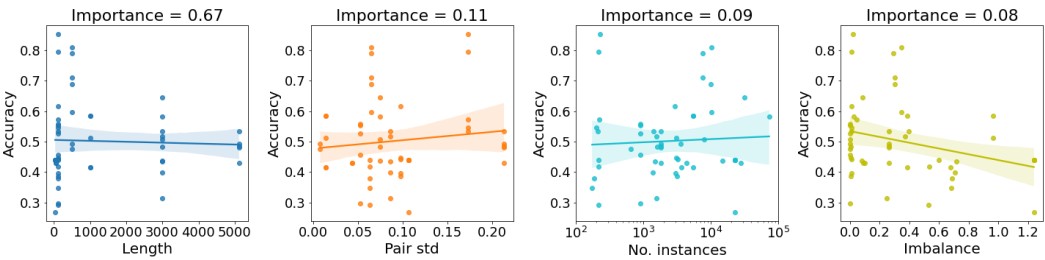

Figure 20: Visualization of OTDA performance versus its four more important data characteristics returned from XGBoost.

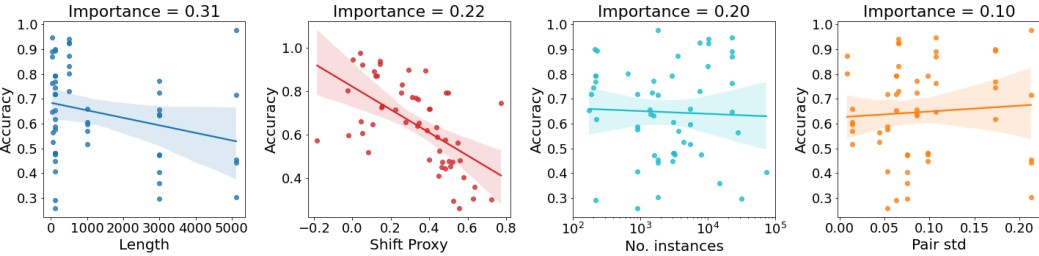

Figure 21: Visualization of Raincoat performance versus its four more important data characteristics returned from XGBoost.

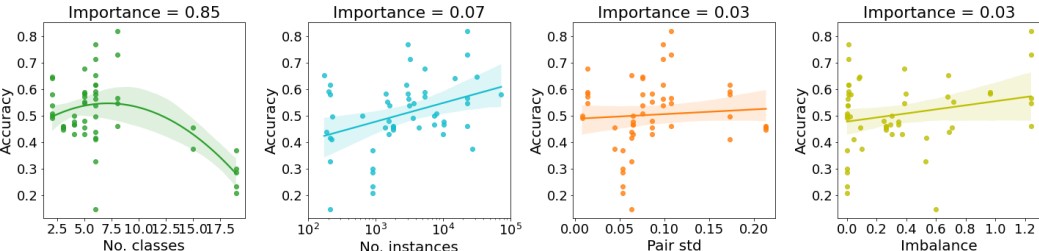

Figure 22: Visualization of VRADA performance versus its four more important data characteristics returned from XGBoost.

The algorithm VRADA implements adversarial learning and uses a VRNN as a backbone. The learning objective of a VRNN is to minimize the distance in distribution between the inference model generated by the VRNN and a prior and to minimize the log-likelihood of the reconstructed input. The VRNN loss is a function of the VRNN itself and we use $\mathcal{L}_{\text{VRNN}}$ to refer to it. The total loss for VRADA is

$$\mathcal{L}_{\text{total}}(E, C, D) = \mathcal{L}_{\text{VRNN}}(E) + \mathcal{L}_C(E, C) - \lambda \mathcal{L}_A(E, D) \,. \tag{4}$$

This objective is minimized with respect to the classifier $C$ and encoder $E$ and maximized with respect to the discriminator $D$.

The Conditional DANN (CDANN) algorithm (Long et al. (2018)) was originally proposed for domain adaptation on images. Basen on DANN, CDANN is also tries to enforce domain adaptation through adversarial learning. In this work we propose the InceptionCDAN algorithm, adapting CDANN to the problem of TSC. In addition to the backbone $E$, classifier $C$, and discriminator $D$, CDANN requires a multilinear map $T : \mathcal{Z} \times \mathcal{Y} \to \mathcal{Z}'$, which combines the latent space representation produced by the backbone with the soft probabilities output by the classifier, thus conditioning the input of the discriminator on the predicted class, with $\mathcal{Z}'$ the input space of $D$.

With the multilinear map, the adversarial loss becomes,

$$\mathcal{L}_A(E, C, D) = \frac{1}{n_S} \sum_{i=1}^{n_S} \mathcal{L}\left( D\left( T\left( E(\mathbf{X}^s), C(E(\mathbf{X}^s)) \right) \right), \mathbf{y}_{\text{domain}}^s \right) \tag{5}$$

$$+ \frac{1}{n_T} \sum_{i=1}^{n_T} \mathcal{L}\left( D\left( T\left( E(\mathbf{X}^s), C(E(\mathbf{X}^s)) \right) \right), \mathbf{y}_{\text{domain}}^t \right) \,. \tag{6}$$

The optimization problem is the same as for the general adversarial domain adaptation framework, with the addition of the multilinear map. Note that $T$ has no learnable parameters.

**Contrastive Learning**  Contrastive learning tries to align the predictions made by the model for pairs of samples coming from two different domains. The CoTMix algorithm computes the Class-aware contrastive loss, $\mathcal{L}_{\text{CAC}}$, to align the predictions between source and source dominant samples from the same class, and the unsupervised contrastive loss $\mathcal{L}_{\text{UC}}$ to align the predictions between target and target dominant samples.

Given $n_S$, the total number of source domain samples, $n_S$ source dominant samples are generated through the temporal mixup operation, making a total of $2n_S$ samples. Let the output probabilities of source domain samples be $\mathbf{o}_i^s = C(E(\mathbf{X}_i^s))$ for any $i \in [n_S]$ and let the output probabilities of source dominant samples be $\mathbf{o}_i^{sd} = C(E(\mathbf{X}_i^{sd}))$ for any $i \in [n_S]$, with encoder $E$ and classifier $C$. Let the overall source samples be $\{(\mathbf{X}_i^{so}, \mathbf{y}_i^{so})\}_{i=1}^{2n_S}$, and their corresponding output probabilities be $\{\mathbf{o}_i^{so}\}_{i=1}^{2n_S}$, assuming that the class label is the same for any two corresponding samples from both domains, i.e. $\mathbf{y}_i^s = \mathbf{y}_i^{sd}$ for any $i \in [n_S]$. Let $U(k) = \{u \in [2n_S] \setminus k : \mathbf{y}_u^{so} = \mathbf{y}_k^{so}\}$ be the set of indices different than $k$ such that their corresponding labels are the equal to $\mathbf{y}_k^{so}$. Thus, the class-aware contrastive loss tries to align the prediction for all source (domain and dominant) samples belonging to the same class.

$$\mathcal{L}_{\text{CAC}}(E, C) = \sum_{k=1}^{2n_S} \frac{-1}{|U(k)|} \sum_{u \in U(k)} \log \frac{\exp\left(\mathbf{o}_k^{so} \cdot \mathbf{o}_u^{so} / \tau\right)}{\sum_{a \neq k} \exp\left(\mathbf{o}_k^{so} \cdot \mathbf{o}_a^{so} / \tau\right)} \,, \tag{7}$$

where the symbol $\cdot$ denotes inner product, $\tau$ is a temperature parameter, and $|U(k)|$ is the cardinality of $U(k)$.

In the target domain, no information is available about the class labels. Thus, contrastive learning can only be done in an unsupervised manner. Given $n_T$, the total number of target domain samples, $n_T$ target dominant samples are generated through the temporal mixup operation, making a total of $2n_T$ overall target samples. Let the output probabilities of target domain samples be $\mathbf{o}_i^t = C(E(\mathbf{X}_i^t))$ for any $i \in [n_T]$ and let the output probabilities of target dominant samples be $\mathbf{o}_i^{td} = C(E(\mathbf{X}_i^{td}))$ for any $i \in [n_T]$. The overall target samples are $\{\mathbf{X}_i^{to}\}_{i=1}^{2n_S}$, and their corresponding output probabilities are $\{\mathbf{o}_i^{to}\}_{i=1}^{2n_S}$, forming a set such that $\mathbf{o}_i^{to} = \mathbf{o}_i^t$ if $i \leq n_T$ and $\mathbf{o}_i^{to} = \mathbf{o}_i^{td}$ otherwise. Define $f(k)$ as the index positive pair of $k$, such that for any $k \in [2n_T]$ $f(k) = k + n_T$ if $k \leq n_T$, and $f(k) = k - n_T$ otherwise. Thus, the unsupervised contrastive loss tries to align the predictions

for pairs of samples, assuming that each target domain sample belongs to the sample class as its corresponding target dominant sample,

$$\mathcal{L}_{\mathrm{UC}}(E, C) = \frac{-1}{2n_T} \sum_{k=1}^{2n_S} \log \frac{\exp\left(\mathbf{o}_k^{so} \cdot \mathbf{o}_{f(k)}^{so}/\tau\right)}{\sum_{a \neq k} \exp\left(\mathbf{o}_k^{so} \cdot \mathbf{o}_a^{so}/\tau\right)} \ . \tag{8}$$

The overall contrastive loss $\mathcal{L}_{\mathrm{contrastive}}$ is the sum of $\mathcal{L}_{\mathrm{CAC}}(E, C)$ and $\mathcal{L}_{\mathrm{UC}}(E, C)$. Additionally, the CoTMix learning algorithm minimizes the cross-entropy loss over source domain samples, $\mathcal{L}_C$, and the entropy over target domain samples,

$$H(E, C) = -\frac{1}{n_T} \sum_{i=1}^{n_T} \mathbf{o}_i^t \cdot \log \mathbf{o}_i^t \ . \tag{9}$$

The total loss for the contrastive learning framework is,

$$\mathcal{L}_{\mathrm{total}}(E, C) = \mathcal{L}_C(E, C) + H(E, C) + \lambda \mathcal{L}_{\mathrm{contrastive}}(E, C) \ , \tag{10}$$

where $\lambda$ is a trade-off between the classification and the contrastive loss functions. The loss function 10 is the learning objective used by the CoTMix and InceptionMix models in our framework.

**Frenquency domain analysis**  One of the main contributions from He et al. (2023) is to analyse time and frequency domain features separately. This allows the model to learn shifts in the distribution of frequency features that might be vital for the main classification task. To this purpose, the Raincoat model uses a frequency encoder, $E_F : \mathcal{X} \rightarrow \mathcal{Z}_F$, which transforms the input into frequency features, with $\mathcal{Z}_F$ the frequency feature space.

On the other hand, time features are extracted by a traditional backbone, such as a 1D CNN in the case of Raincoat or the Inception backbone in the case of InceptionRain. We denote the time encoder $E_T : \mathcal{X} \rightarrow \mathcal{Z}_T$, with $\mathcal{Z}_F$ the time feature space. The output of the overall encoder is the concatenation of the ouputs of the time and frequency encoders, $E(\mathbf{X}) = (E_T(\mathbf{X}), E_F(\mathbf{X}))$, for any $\mathbf{X} \sim \mathcal{X}$.

The Raincoat algorithm aims to align the latent space representations of source and target domain samples. Let $\mathbf{Z}^S = \{E(\mathbf{X}) : \mathbf{X} \in D_S\}$ be the set of latent space representations of samples in the source domain, and similarly for the target domain let $\mathbf{Z}^T = \{E(\mathbf{X}) : \mathbf{X} \in D_T\}$. The Sinkhorn divergence is computed between the sets $\mathbf{Z}^S$ and $\mathbf{Z}^T$ and minimized with respect to $E$. Thus, $\mathcal{L}_{\mathrm{Sinkhorn}}(E) = \mathrm{Sinkhorn}(\mathbf{Z}^S, \mathbf{Z}^T)$.

Additionally, Raincoat promotes learning of semantic features by minimizing a reconstruction loss. Let $G : \mathcal{Z} \rightarrow \mathcal{X}$ be a decoder function that reconstructs the input from its latent space representation. The reconstruction loss is computed between input samples and their reconstructed counterparts, and it is optimized with respect to the encoder $E$ and the decoder $G$. Thus, $\mathcal{L}_R(E, G) = \frac{1}{n_S} \sum_{i=1}^{n_S} d(\mathbf{X}_i^s, G(E(\mathbf{X}_i^s))) + \frac{1}{n_T} \sum_{i=1}^{n_T} d(\mathbf{X}_i^t, G(E(\mathbf{X}_i^t)))$, with $d$ some distance in $\mathcal{X}$. The total loss function used in the Raincoat algorithm is,

$$\mathcal{L}_{\mathrm{total}}(E, G, C) = \mathcal{L}_C(E, C) + \mathcal{L}_{\mathrm{Sinkhorn}}(E) + \mathcal{L}_R(E, G) \ . \tag{11}$$

## A.4  ADDITIONAL UDA CLASSIFIER: SASA

In this section we explore the Sparse Associative Structure Alignment (SASA) model published in Cai et al. (2021). The latter algorithm uses the sparse associative structure discovery method coupled with an adaptive summarization of subsequences extracted from the original series and a final Maximum Mean Discrepancy (MMD) based structure alignment method to transfer the knowledge from the source domain to the target domain. The default backbone for extracting features from subsequences is a single layer LSTM. The results depicted in Figure 23 show that SASA occupies the lowest average rank amongst all classifiers which is expected and can be explained by various

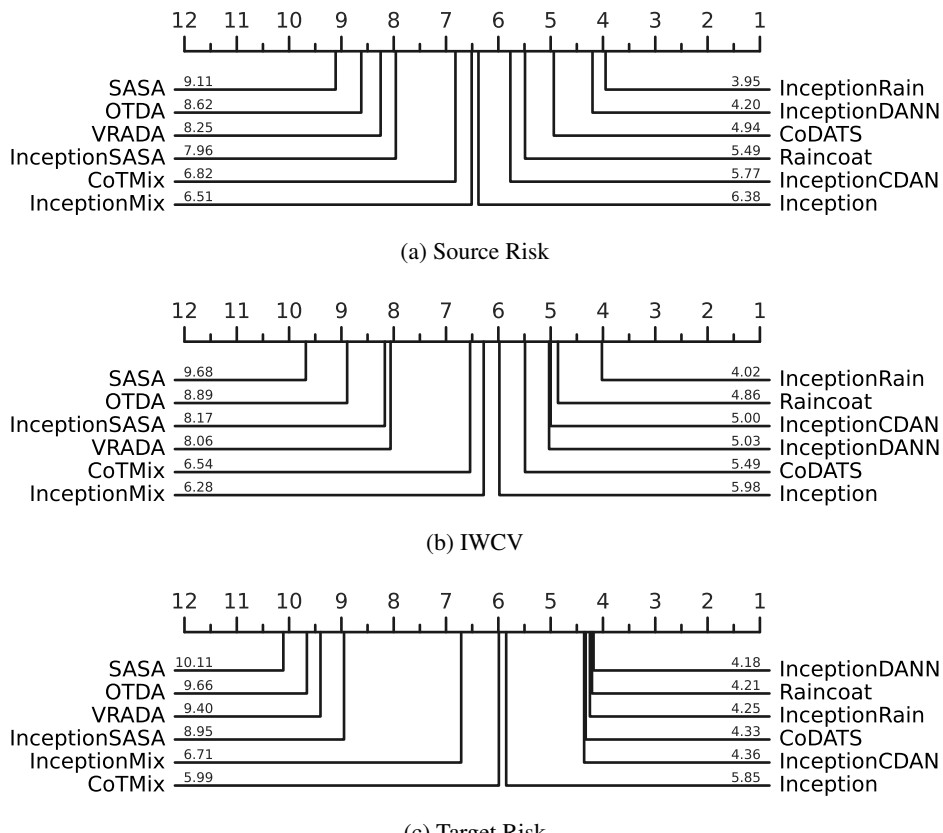

(a) Source Risk

(b) IWCV

(c) Target Risk

Figure 23: Average rank diagrams based on accuracy for different hyperparameter tuning methods with the Sparse Associative Structure Alignment (SASA) model added to the mix of classifiers

reasons. First, the model is originally designed for detecting events in a time series rather than classifying a whole time series as input, which explains the author's approach of taking into consideration solely the last ($\tau < length$) time steps when generating the subsequences while disregarding the earlier time steps. Given that the task of TSC does not assume any prior on the importance of different time stamps, the task seems to be inherently different for SASA. For example for a very long time series such as in those present in the SleepStage dataset, SASA might drop more than $90\%$ of the whole series and use the last $10\%$ for classification. Nevertheless we have implemented and included the model here in an extension of the benchmark depicted in Figure 23, but rather removed it from the original benchmark in the main paper given the tasks' dissimilarities. Finally we should note that the backbone used by SASA is a recurrent neural network which is well adapted given the aforementioned task of event detection, however not very well suited for traditional TSC (Ismail Fawaz et al., 2019). The latter claim is further validated when comparing the rank of InceptionSASA to SASA, clearly substituting the LSTM backbone with an Inception backbone gives a boost in performance, yet not enough to overcome the average methods for UDA that fully utilize the complete input time series.

