# OpenReview forum: "DEEP UNSUPERVISED DOMAIN ADAPTATION FOR TIME SERIES CLASSIFICATION: A BENCHMARK"
_ICLR.cc/2024/Conference — Submitted to ICLR 2024_

### Official Review · Reviewer_wzDS · 2023-10-26

**Soundness:** 3 good
**Presentation:** 3 good
**Contribution:** 4 excellent
**Rating:** 6
**Confidence:** 3

**Summary:**

This work explores the use of unsupervised domain adaptation (UDA) for time series classification (TSC), with a particular focus on deep learning methods. In UDA, which has been extensively explored in vision and natural language applications, two domains of data exist: a labelled source domain and an unlabelled target domain that has some form of shift in the time series data (e.g. differences in data used for training and data used during deployment). The objective is to leverage the labelled source data to make predicts in the target domain.

In addition to five existing datasets, this work proposes the use of seven new datasets for UDA TSC (taken from existing sources). This collection of datasets serves as a benchmarking evaluation tool for assessing the efficacy of different UDA TSC deep learning approaches, notably with different algorithms, hyperparameter optimisation approaches, and model backbones. Consistent experimentation is used to compare the performance of these different approaches and make observations of which elements contribute the most to performance.

**Strengths:**

**Originality**
O1. The main novelty of the work lies in the proposal of additional datasets and a consistent, fair framework for evaluation the UDA TSC methods. This also extends to the insights that can be drawn from this evaluation.
O2. There is also some originality in the deep UDA methods that are used, notably using consistent a consistent backbone (InceptionTime) across different approaches.

**Quality**
Q1. The experimental setup is well-structured and makes steps to ensure fairness across all algorithms (e.g. limiting GPU time for training/hyperopt).
Q2. Results analysis provides some comparisons between the choice of different classifiers, hyperopt methods, and backbones.

**Clarity**
C1. Clear descriptions of all the different elements of the experiments are given (models, hyperopt methods, datasets, and pipelines).
C2. Figures are communicative and support conclusions drawn from the work.

**Significance**
S1. This work could serve as a stable baseline for further developing UDA TSC deep learning approaches, helping to progress the area of research.
S2. Insights into the performance of different methods (e.g. InceptionRain seemingly being the strongest method) is useful for establishing the current SOTA and assessing the relative performance.

**Weaknesses:**

**Presentation of Results**
P1. While Figure 1 compares model performance within hyperopt methods (a, b, c), it does not provide an overall comparison of all models with all hyperopt methods. As such, it is difficult to determine which complete approach (model + tuning approach) actually has the best performance. An additional critical difference diagram comparing the (5?) top methods for each tuning approach would make this much clearer.
P2. While the figures provide some information, and the Appendix gives a full set of results for each dataset, it remains difficult to assess the margins between the approaches. A summary table of average accuracy across all datasets for each experimental configuration would be beneficial in conveying this information.
P3. Further variations of Figure 4 for other models/datasets would be useful to see if the revealed trend is consistent.
P4. I think the violin plots in Figure 7 are a strong way of communicating the results, and potentially should be moved to the main body if possible. As mentioned above, combining the some selection of the top methods for each tuning approach into a single plot would further aid comparison.

**Significance**
S1. I believe this work has the most potential if the evaluation is released to allow further development of methods. I appreciate the source code is planned to be released upon acceptance, but potentially taking this a step further and allowing for easy reproducibility/extensibility would improve the impact of the work and help progress the research area.

**Questions:**

1. To what extent is there dataset imbalance in the datasets? Additional results, for example using balanced accuracy, may be warranted if dataset imbalanced is high. At the very least, a discussion on any dataset imbalances would be helpful. I appreciate F1 score results are given in the appendix.

---

> ### Author Response · Authors · 2023-11-17
>
> Thank you for your valuable comments and suggestions.
> We particularly appreciate all the strengths you mentioned about our work.
>
> Here are our replies to the weaknesses you raised.
>
> **Presentation**
> - P1. We acknowledge the reviewer's suggestion to incorporate an *additional critical difference diagram* for a comprehensive comparison of all experiments, presenting an overall perspective on performance. This point has been addressed in the Appendix A.2.1. The average accuracy rank across all algorithms and model selection methods delineates three clusters: \
> i) the first cluster comprises most of the algorithms trained using the target-only method, \
> ii) a second cluster includes OTDA and VRADA, identified as the less performing algorithms across the three hyperparameter selection methods, and \
> iii) a third cluster encompasses the remaining algorithms, fairly/visually distributed between the source-only and IWCV methods. Due to space constraints, we kept this further analysis in the Appendix only.
> - P2. We have included a comprehensive table, where each column corresponding to a specific method (Source Risk, IWCV, Target Risk), in the Appendix A.1.2. The table aims to provide *a summary by averaging accuracy scores across datasets for each classifier*, accompanied by reporting of standard deviations.
> - P3. We have added *variations of Figure 4* with different datasets and UDA algorithms in Appendix A.2.5, as well as an analysis of these graphs. They confirm a certain correlation of the source risk and IWCV with the target risk, but show a plateau for the source risk, and some different optimum for IWCV, thus explaining the gap in performance compare to target risk.
> -P4. We agree with the reviewer's comment regarding the effectiveness of the plots in Figure 8 for conveying results.
>     Due to space limitations, these additional plots have been placed in the appendix alongside all the extra graphs.
>
> **Significance**
> - S1. We completely agree with this point and we would like to answer with the following elements. First, the results are part of the git repository in a CSV format, and will be available when we open-source it for checking reproducibility and going further in the analysis. Furthermore,  the implementation of our benchmark relies on existing tools to strengthen its *reliability, scalability and modularity* and that we believe will help other research to easily add datasets and/or algorithms. These tools include Hydra for the configuration of the pipelines and parameters, HuggingFace for the datasets and evaluation of the models, PyTorch with HuggingFace, scikit-learn and POT for training the algorithms. As a proof of its modularity, we can mention that it only took us a few hours to add an additional algorithm (SASA) to the benchmark and run the experiment with it during the current rebuttal phase.
>
> **Questions**
> - Q1. We appreciate your highlighting of this analytical aspect. To address this, we have included a new section in the Appendix A.1.3 specifically dedicated to emphasizing the *dataset imbalances*. In this section, we introduced a scoring metric, denoted as I, which quantifies the degree of imbalance, where a score of 0 indicates a highly balanced dataset, and a score of 1 signifies high imbalance. The results are presented in a table format, summarizing the number of classes, samples, and I scores for both the source (train, test) and target (train, test) datasets. It is important to note that only three datasets exhibit significant imbalance, posing additional challenges to the classification task. Note that two out of these three datasets are part of the original established set of datasets for UDA TSC.

---

> > ### Comment · Reviewer_wzDS · 2023-11-20
> > **Follow Up to Rebuttal for Reviewer wzDS**
> >
> > Thank you very much for the response and additional work. The authors have done a good job of addressing my concerns and providing additional figures. I have no further follow up questions.

---

### Official Review · Reviewer_hmFL · 2023-10-30

**Soundness:** 2 fair
**Presentation:** 2 fair
**Contribution:** 1 poor
**Rating:** 3
**Confidence:** 4

**Summary:**

In this paper, the authors present benchmark research on deep unsupervised domain adaptation (UDA) for time series classification (TSC). Specifically, seven new datasets are introduced for this TSC UDA task, and experiments of several existing TSC UDA baselines are tested on these datasets.

**Strengths:**

Strength:

1.	The paper introduces 7 new datasets for TSC UDA task.

2.	The paper conduct experiments on several existing UDA baselines on the new datasets.

3.	The paper has potential to be a benchmark for the following TSC UDA research.

**Weaknesses:**

Weakness:

1.	The major concern of the work is on the technical novelty. All the datasets and baselines (including hyper-parameter tuning methods) are from existing literatures, and there is no novel technical contribution proposed.

2.	For UDA TSC, some important related works are missing, for instance (to name a few), unsupervised video domain adaptation [ref1], transfer gaussian process [ref2], and time-series domain adaptation [ref3]. The authors may need to present a more comprehensive related work section to discuss more related works.

3.	More analyses on the new datasets are expected, for instance, the domain discrepancy analyses (both marginal and conditional can be involved) on different domain pairs. From the experiments results of the source only baseline, the domain discrepancy differs considerably among different domain pairs, see, Table 5 domain 0 and domain 3, Table 9 domain 9 and domain 18.

4.	More analyses on the comparison results are also expected. For instance, analyzing why some baselines achieve positive transfer on some tasks but negative transfer on others, e.g., see OTDA/VRADA in table 9. It would be more interesting to see constructive insights or conclusions that can benefit the community.

[ref1] Video Unsupervised Domain Adaptation with Deep Learning

[ref2] Adaptive Transfer Kernel Learning for Transfer Gaussian Process Regression

[ref3] Time Series Domain Adaptation via Sparse Associative Structure Alignment

**Questions:**

Please refer to the weakness.

---

> ### Author Response · Authors · 2023-11-17
>
> Thank you for your valuable comments and suggestions. Here are our replies to the weaknesses you raised:
> 1. We agree with the reviewer that the technical novelty is limited, but we think this is expected from any benchmark study, as the goal is to assess the quality of existing algorithms on data which should be available openly to the community. Still, we have not limited ourselves to existing TSC UDA dataset and we adapt 7 diverse time series datasets to UDA settings which have never been used before for UDA. Moreover, we have not limited ourselves to simply comparing existing methods, as this could be expected from a benchmark, but we proposed the usage of the Inception backbone that, not only achieves the best performance, but also allows a fair comparison between SOTA methods having different backbones.
> 2. We thank the reviewer for pointing out these references. We believe the work on  UDA for video data [ref1] is a particular field and should not be included here, as it is a combination of image (spatial) and time series (temporal) problems and therefore goes beyond the scope of our study. Nevertheless we have added the latter paper as part of our references of UDA for computer vision tasks.
>     Furthermore regarding the Gaussian Process Regression [ref2], although the paper tackles time series and tabular data, it focuses on regression (data extrapolation) rather than classification tasks and does not tackle the unsupervised DA setting. Nevertheless we have now mentioned regression DA task in the introduction section.
>     Finally, regarding the SASA model [ref3], we strongly agree with the reviewer that this reference is relevant to our benchmark. We therefore have added the Section A.4 in the appendix where we present the results of SASA (with both LSTM and Inception backbones) over all 54 datasets (accounting to almost 1296 hours of gpu time during the current rebuttal phase). However, we decided to keep it outside the main benchmark as it is more suited for the task of detecting an event within a time series, and not classifying the complete input time series, since it is making the decision based on the last few time stamps, which probably explains its bad performance in our study. Comments were also added in the main paper to mention SASA.
> 3. We strongly acknowledge the significance of quantifying the shift between various source and domain pairs, as it holds the potential to offer valuable insights into when domain adaptation proves beneficial and which algorithm is most effective. However, given that quantifying the shift in DA is already difficult, adding to that the complex dynamics of a time series data, we believe that this domain shift quantification is an ongoing area of research that we plan to address in our future work. The complexities stem from the unique nature of time series data, where shifts can manifest in time, features, or a combination of both. Measuring domain discrepancy, e.g. through metrics like Maximum Mean Discrepancy or Wasserstein distance, on different domain pairs may not provide sufficient information about time series shift. As an illustration of this difficulty, OTDA leverages this Wasserstein distance, which quantifies the shift, but does not work well for time series data in our benchmark.
>     We found that using the performance of Inception (i.e. with no adaptation) was a quick and easy proxy to compute the shift, which can be seen in Figure 3, although we cannot assess how good a proxy it is.
>     We also added a paragraph in Appendix A.1.1 to explain some of the discrepancies for specific cases (e.g. when we know that one domain is more noisy than others).
> 4. We believe that there is already some interesting insight for the community even if they are indeed quite general. We cannot really enter too much into details of combination of algorithm/dataset.\
>   i. The limited impact of IWCV compared to Source Risk, which is now additionally supported by the added Figure 6 that mix the methods.\
>   ii. The limited impact of backbone, additionally supported by the comparison for different dataset theme and method.\
>   iii. The fair comparison between algorithms with the same common Inception backbone, and the good performance of InceptionRain.\
>     We have also added some more analysis in the Appendix following the other comments.

---

> > ### Comment · Reviewer_hmFL · 2023-11-20
> > **Thanks for the authors' response.**
> >
> > I really appreciate the authors' reponse to my comments. However, I am still not convinced.
> >
> > Regarding point 1, as a paper submitted to the technical track of ICLR, technical contribution of the paper is the key point to be evaluated. The reviewer suggests the authors to conisder some review/benchmark specific tracks which are more matched with the content of the current paper.
> >
> > Regarding point 2, I am not sure how the authors define the "time series" data as video data is definitly temproal data although it contains more spatial information. Video UDA is also a popular research problem. I highly suggest the authors to include this research line into their work for a more comprehensive benchmark study. [ref2] can be easily adapted to classification as regression is actually a more general task. I agree that it is not specific to UDA setting, but i encourage the authors to have a discussion in related work. Moreover, unsupervised domain adpatation regression is a clear related research line that should be included.
> >
> > Regarding point 3, the reviewer agrees that quantifying the domain shift of temporal domains is not easy, and thus it would be a good point to increase the technical value of the paper. I highly encourage the authors to work out this point which may not only introduce more technical contributions but also support the emprical analyses.
> >
> > Lastly, I agree the paper contains interesting points as I stated in the strength. However, I also have some concerns which lead me to the current score. I would suggest the authors to consider some review/benchmark tracks with my comments taken into account in the revision.

---

> > > ### Author Response · Authors · 2023-11-22
> > >
> > > We would like to thank the reviewer again for taking their time in checking and replying to our comments.
> > >
> > > As it is the major concern for the reviewer, we would like to point out that we have submitted the paper to ICLR with the primary area "datasets & benchmarks".
> > > We should note that to the best of our knowledge ICLR does accept benchmark papers such as [ref1,ref2] to cite a few; where decisions were taken with the following reviewers' remarks _"a useful benchmark for deep learning optimizers, but limited research contribution"_ and _"The paper was discussed by the reviewers after the authors correctly pointed out that methodological machine learning novelty is not a necessary condition for accepting papers"_. We hope that this will be taken into account for the final evaluation of our paper.
> > >
> > > [ref1] Schneider, F., Balles, L., & Hennig, P. (2018, September). DeepOBS: A Deep Learning Optimizer Benchmark Suite. In International Conference on Learning Representations.
> > > [ref2] Mehrjou, A., Soleymani, A., Jesson, A., Notin, P., Gal, Y., Bauer, S., & Schwab, P. (2021, October). GeneDisco: A Benchmark for Experimental Design in Drug Discovery. In International Conference on Learning Representations.

---

### Official Review · Reviewer_KuTn · 2023-11-01

**Soundness:** 2 fair
**Presentation:** 2 fair
**Contribution:** 2 fair
**Rating:** 5
**Confidence:** 3

**Summary:**

This paper presents a thorough benchmarking study on time-series unsupervised domain adaptation, primarily focusing on deep learning techniques. It examines the impact of model backbones and hyperparameter tuning approaches. Furthermore, the authors evaluate various existing unsupervised domain adaptation methods across multiple domains, including seven new benchmark datasets.

**Strengths:**

1. The study delivers a detailed benchmark on unsupervised domain adaptation for time-series data, delving into the effect of domain adaptation algorithms, model backbones, and hyperparameter tuning strategies.
2. The paper evaluates a range of unsupervised domain adaptation methods on datasets from diverse domains, including seven newly introduced datasets and existing benchmarks.

**Weaknesses:**

1. The discussion on the effect of model backbone in the paper is limited primarily to the Inception model. A broader examination involving diverse backbone models is crucial to substantiate the claim that "backbones do not have a significant impact".
2. More discussions on different types of unsupervised domain adaptation methods would be beneficial. Specifically, it would be informative to explore under what specific conditions certain domain adaptation approaches may outperform others.
3. Additional discussions regarding the choice of model backbones is helpful too. For example, I am curious if Inception is the best model backbone over all domains, or we need different model backbones for different time-series domains and data characteristics.
4. Figure 2 (b) and 5 should be merged since they lead to similar findings.

**Questions:**

See Weaknesses above.

---

> ### Author Response · Authors · 2023-11-17
>
> Thank you for your valuable comments and suggestions. Here are our replies to the weaknesses you raised:
>  1. Note that we compare Inception with 3 different architectures that are proposed in the literature (CoDATS, Raincoat and CoTMix). Because it is hard to compare the different backbones of different methods directly, we chose the Inception backbone as reference. Following your remark, we added a section (Appendix 2.6) and three plots in Figure 11, that compare the 2 additional backbones individually with Inception. The same conclusion holds for each of them. Furthermore With this result, we believe that there is enough argument for the limited impact of backbone in time series UDA. Additionally, a proper comparison of another classical backbone would lead to around 5 months of GPU’s time.
> 2. We agree with the reviewer about the necessity of a deeper analysis. However, we believe part of this analysis relies on an estimation of the shift between the datasets to understand how much shift in the data each method can support. The estimation of the shift being still an open issue and difficult to tackle, the best we proposed here is an analysis about the difficulty of a dataset, which we can assess with accuracy scores as well as standard deviation from pairs of source-target.
> 3. Firstly, the InceptionTime algorithm has been found to be the best deep learning algorithm in a recent benchmark for Time Series Classification [ref1], which is the reason why we chose this backbone for a common comparison of the UDA approaches.
>     Following your comment, we added in Section A.2.6 the Figure 12 to provide a better comparison about the differences between various types of dataset (Machinery, Medical, Motion and Remote sensing) and add comments, especially on a potential bias toward Motion dataset for existing methods due to existing and already established UDA TSC datasets. The latter observation justifies the need to expand the list of UDA TSC datasets and improve generalization to different TSC domains, which is one of the main contributions of this paper.
> 4. Indeed, both figures have similar finding and can be merged but we still think that focusing specifically on Raincoat, which is the best method, is still important to support our claim: it is the Raincoat’s UDA method behind the better performance rather than its backbone.
>
> [ref1]: Ruiz, A.P., Flynn, M., Large, J. et al. The great multivariate time series classification bake off: a review and experimental evaluation of recent algorithmic advances. Data Min Knowl Disc 35, 401–449 (2021)

---

> > ### Comment · Reviewer_KuTn · 2023-11-21
> >
> > Thank the authors' efforts for the response and additional experiments. I still have some concerns after reading the rebuttal.
> >
> > 1. Model Type Comparisons: The current analysis incorporates three to four specific models, which provides a valuable starting point. However, for a benchmark study, more comprehensive comparisons across different model types would be better. For example, how do CNN, RNN, Transformer based models compare with each other? How does the depth of the network affect performance?
> > 2. Analysis of UDA Methods Selection: I hope to see some analysis on the selection of UDA methods. For example, it would be great if we could have a figure that illustrates the performance of different UDA methods under varying degrees of distribution shift from source to target.
> > 3. The author wrote "we added a section (Appendix 2.6) and three plots in Figure 11", but I only see two plots in Figure 11.

---

> > > ### Author Response · Authors · 2023-11-23
> > >
> > > We would like to thank the reviewer again for taking their time in checking and replying to our comments.
> > >
> > > 1. **Impact of depth/other hyperparameters**: Given that we performed a random search, all hyperparameters were varied simultaneously thus we cannot perform a sensitivity analysis on one parameter only. Nevertheless we have checked the depth impact but could not reach any clear conclusion worth mentioning. **Comparison of models**: Due to the resource and computational limitations, we chose to use the current best architecture (Inception [ref1]) for time series classification as a common backbone. RNN have been shown to perform worse than CNN for time series classification [ref2], and there are only few works with transformers in that context [ref1].
> > > 2. We added a **high-level analysis** in Appendix A.2.7 (text in dark red color) showing how some data characteristics affect the performance of each UDA algorithm. In particular, we provide plots showing how they are affected by the shift, imbalance, number of classes, etc.
> > > For instance, Raincoat's most important feature is found to be the length of the time series (see Figure 21), which is consistent with its "step: 1 alignment of the time series" implementation. The reason behind this lies in the architecture of the proposed CNN, which relies on a fixed kernel size. When using the inception backbone instead, this feature is not important anymore, as Inception is not really impacted by the length by design since it proposes different kernel sizes in parallel to be adapted to the time series length.
> > > 3. We apologize for the error. We had removed one of the 3 plots because it was already in the main paper and we forgot to modify our answer.
> > >
> > > [ref1] Ruiz, A.P., Flynn, M., Large, J. et al. The great multivariate time series classification bake off: a review and experimental evaluation of recent algorithmic advances. Data Min Knowl Disc 35, 401–449 (2021). https://doi.org/10.1007/s10618-020-00727-3
> > >
> > > [ref2] Smirnov, D., & Nguifo, E. M. (2018). Time series classification with recurrent neural networks. Advanced analytics and learning on temporal data, 8.

---

### Author Response · Authors · 2023-11-17
**Overall Comments**

We would thank all reviewers for their time and valuable feedback. In the following, we consolidate and address the key points raised in the reviews and discussions. Additionally, we provide detailed answers to each reviewer in separate comments.
We are particularly grateful for the reviewers' appreciation of our benchmarking work and its relevance to the community.

The reviewers recommended a more thorough analysis of the results to show different insights of our experiments. In response, we have incorporated multiple tables and figures in various sections of the Appendix, accompanied by corresponding analyses. We conducted a comparison of backbones for each dataset theme (Medical, Remote Sensing, etc.). Furthermore, we included a high-level analysis illustrating the impact of certain metadata characteristics on the performance of each UDA algorithm. Additionally, we introduced and assessed an additional classifier, the Sparse Associative Structure Alignment (SASA) model.

We would like also to mention hmFL's comment regarding novelty, emphasizing it as a significant concern. We would like to point out that we have submitted our paper with primary area "datasets & benchmarks", where limited novelty is commonly anticipated in benchmark studies. Furthermore, we believe that our contribution goes beyond a mere comparison of existing methods and datasets. Notably, we introduced a new Inception backbone to facilitate a fair comparison of various UDA methods.

---

### Meta-Review · Area_Chair_5TXH · 2023-12-04

**Metareview:**

This paper was reviewed by three experts and received 5, 3 and 6 as the ratings. It was extensively discussed between the authors and the reviewers during the rebuttal period. The reviewers acknowledged that the paper evaluates a range of unsupervised domain adaptation methods on datasets from diverse domains, including seven newly introduced datasets. However, the reviewers raised concerns about the lack of comprehensive experimental evaluations. It was mentioned that more detailed analysis (with additional backbones and other baseline methods, as well as domain discrepancy analysis of the new datasets) and discussions are needed.

We appreciate the authors' efforts in meticulously responding to each reviewer’s comments. We also appreciate the authors' efforts in conducting additional experiments with more backbones besides Inception, to address the concerns of reviewer KuTn, and also in updating their paper to address the concerns about presentation and data imbalance of reviewer wzDS. However, during the AC-reviewer discussion period, the reviewers shared a common concern about the lack of comprehensive evaluation of the proposed method, and agreed that more in-depth analysis is required with more backbones and other baseline methods to publish this research. In light of the above discussions, we conclude that the paper may not be ready for an ICLR publication in its current form. While the paper clearly has merit, the decision is not to recommend acceptance.

A paper submitted to the “datasets and benchmarks” track should be structured like a usual research paper, where the dataset collection part should be described as a research problem on how the authors addressed the difficulties and challenges faced, the feature engineering part should be described as a research problem on how alternative choices were studied and the benchmarking part should be described as a research problem with some conclusions and insights for future practitioners. We hope these suggestions are helpful and encourage the authors to consider the reviewers’ comments when revising the paper for submission elsewhere.

**Justification For Why Not Higher Score:**

All the reviewers agreed that the paper needs more in-depth experimental analysis before it can be accepted for publication in a reputed conference like ICLR.

**Justification For Why Not Lower Score:**

N/A.

---

### Decision · Program_Chairs · 2024-01-16

Reject